

# Assessing the dependence structure between oceanographic, fluvial, and pluvial flooding drivers along the United States coastline

Ahmed A. Nasr[1], Thomas Wahl[1], Md Mamunur Rashid[1], Paula Camus[2], Ivan D. Haigh[2]

[1]Civil, Environmental, and Construction Engineering & National Center for Integrated Coastal Research, University of Central Florida, 12800 Pegasus Drive, Suite 211, Orlando, FL 32816-2450, USA
[2]School of Ocean and Earth Science, National Oceanography Centre Southampton, University of Southampton, Waterfront Campus, European Way, Southampton, SO14 3ZH, UK

*Correspondence to*: Ahmed A. Nasr (ahmed.nasr@knights.ucf.edu)

**Abstract.** Flooding is of particular concern in low-lying coastal zones that are prone to flooding impacts from multiple drivers: oceanographic (storm surge and wave), fluvial (excessive river discharge), and/or pluvial (surface runoff). In this study, we analyse for the first time the compound flooding potential along the contiguous United States (CONUS) coastline from all flooding drivers, using observations and reanalysis datasets. We assess the overall dependence from observations by using Kendall's rank correlation coefficient ($\tau$) and tail (extremal) dependence ($\chi$). Geographically, we find highest dependence between different drivers at locations in the Gulf of Mexico, southeast, and southwest coasts. Regarding different driver combinations, the highest dependence exists between surge-waves, followed by surge-precipitation, surge-discharge, waves-precipitation, and waves-discharge. We also perform a seasonal dependence analysis (tropical vs extra-tropical season), where we find higher dependence between drivers during the tropical season along the Gulf and parts of the East coast and stronger dependence during the extra-tropical season on the West coast. Finally, we compare the dependence structure of different combinations of flooding drivers using observations and reanalysis data and use the Kullback–Leibler (KL) Divergence to assess significance in the differences of the tail dependence structure. We find, for example, that models underestimate the tail dependence between surge-discharge on the East and West coasts and overestimate tail dependence between surge-precipitation on the East coast, while they underestimate it on the West coast. The comprehensive analysis presented here provides new insights on where compound flooding potential is relatively higher, which variable combinations are most likely to lead to compounding effects, during which time of the year (tropical versus extra-tropical season) compound flooding is more likely to occur, and how well reanalysis data captures the dependence structure between the different flooding drivers.

## 1 Introduction

The Contiguous United States (CONUS) comprises 48 states (i.e., all states excluding Hawaii and Alaska). Approximately 40% of the United States (US) population lives in coastal counties which make up less than 10% of the total area of the CONUS; this leads to a high population density relative to inland areas, especially in the 17 major port cities with over 1 million inhabitants located along the US coast (Hanson et al., 2011). The coastal counties combined, if they were a single





country, would rank third in the world in terms of the gross domestic product (GDP) after the US and China (NOAA Office for Coastal Management, 2021). Furthermore, 40% of the people living in coastal counties are at high risk of being affected by coastal flood hazards, including vulnerable populations such as elderlies, children, non-native English speakers, and the poor (NOAA Office for Coastal Management, 2021).

Floods are the most dangerous and costly natural disaster. In the US, the total direct economic losses from major weather and climate disasters (where each disaster caused a minimum direct loss of USD 1 billion) amounted to USD 1.75 trillion for the period 1980-2020 (Smith, 2020). 66% of these losses (USD 1.15 trillion) resulted from inland floods (33 events) and tropical cyclones (52 events) causing extreme wind, rain, storm surge, and waves. Hurricanes Harvey in 2017 and Katrina in 2005 both had estimated damages totalling around USD 300 billion (Smith, 2020). In low-lying coastal areas flooding occurs due to

different meteorological and hydrological drivers, including: storm surge and waves (both oceanographic), excessive river discharge (fluvial), and direct runoff due to precipitation (pluvial). Impacts from these four drivers can be exacerbated depending on local characteristics if they occur concurrently (at the same time) or in close succession (separated by a few hours or days), a phenomenon that is known as 'compound flooding'.

The definition of compound events has evolved over the past decade (e.g., Seneviratne et al., 2012; Leonard et al., 2014;

Zscheischler et al., 2018). A widely adopted definition is the one by Zscheischler et al. (2018) defining compound events as "a combination of multiple drivers and/or hazards that contributes to societal or environmental risk". Compound meteorological and hydrological extremes have received increased attention due to their adverse impacts on the environment, society, and economy. Flood risk assessments (including those conducted in coastal locations) traditionally account for individual drivers and independence between them is often falsely assumed, which can lead to an underestimation of flood

risk (Wahl et al, 2015).

According to the proposed typology in Zscheischler et al. (2020), compound flooding is considered as a multivariate event where multiple climate drivers and/or hazards can occur in the same geographical region, that may not be extreme themselves, but their joint occurrence leads to extreme impacts. The four main flooding drivers in coastal regions are often causally related through associated weather patterns; for instance, when a storm causes extreme rainfall, storm surge and/or high waves, and

river discharge is enhanced by local characteristics of the catchment (Hendry et al., 2019). The statistical modelling framework suggested by Zscheischler et al. (2020) for this type of multivariate compound event consists of multivariate probability distribution functions, which represent both the marginal distributions and dependence of multiple random variables. High-dimensional datasets can be modelled using copula-based approaches, but due to their complexity these multivariate statistical models have mostly been applied in local studies (Lian et al., 2013 in Fuzhou China; Kew et al., 2013 in the Netherlands;

Rueda et al., 2016 in Santander, Spain; Bevacqua et al., 2017 in Ravenna, Italy; Couasnon et al., 2018 in Houston, US; Jane et al. 2020 in South Florida, US; Santos et al., 2021 in Texas, US). At larger spatial scales (continental to global), where the compound flooding risk varies along coastlines, previous assessments were often limited to the bivariate case where two flooding drivers were analysed (e.g., Zheng et al., 2013; Wahl et al., 2015; Moftakhari et al., 2017; Paprotny et al., 2018; Ward et al., 2018; Marcos et al, 2019; Hendry et al, 2019; Couasnon et al., 2020). There are some notable exceptions where





dependence between three or even all four flooding drivers were considered, but those focused only on Europe (Petroliagkis et al., 2016; Paprotny et al., 2020; Camus et al., 2021). At the global scale, Bevacqua et al. (2020) quantified the dependence between sea level and discharge and sea level and precipitation to explore if one is a reasonable proxy for the other, and under which conditions. In addition, Ridder at al. (2020) identified hotspots and assessed the statistical dependence for different combinations of hazards and hazard drivers, including coastal flooding drivers.

Typical flooding driver combinations that were previously assessed include: surge and discharge (e.g., Moftakhari et al., 2017); surge and precipitation (e.g., Wahl et al., 2015); surge and waves (e.g., Marcos et al., 2019); surge, discharge, and precipitation (e.g., Svensson and Jones, 2002; 2004); surge, waves, and discharge (e.g., Petroliagkis et al., 2016); and surge, waves, discharge, and precipitation (e.g., Hawkes and Svensson, 2006; Camus et al., 2021). Many studies were performed using observational data (e.g., Wahl et al., 2015; Ward et al., 2018), while some included model hindcast data (Marcos et al., 2019;

Couasnon et al., 2020; Camus et al., 2021), and very few included both or compared different datasets (e.g., Paprotny et al., 2020; Ganguli et al., 2020; Zscheischler et al., 2021). For the CONUS coastline, two previous studies assessed compound flooding potential at the continental scale (while the CONUS was also included in global scale assessments): Wahl et al. (2015) analysed storm surge and precipitation and Moftakhari et al. (2017) analysed storm surge and discharge. Both studies highlighted that existing dependence between coastal and freshwater flooding drivers should be taken into account for coastal

flood risk assessments and that non-stationarity can lead to a further increase in compound flooding potential.

Here we build on these previous studies and perform the first continental-scale analysis of the compound flooding potential caused by oceanographic (storm surge and waves), fluvial (excessive river discharge), and pluvial (direct surface runoff) sources using both observational and model hindcast/reanalysis data. We have three key objectives. Our first objective is to characterize and map the dependence between different drivers at locations around the CONUS coastline and identify spatial

patterns. We carry out this specific objective using different methods to quantify the (bivariate) dependence between the variables representing the flooding drivers. This will show where compound flooding potential is relatively higher and which pairs of drivers are more likely to lead to compounding effects. Our second objective is to perform the dependence analysis separately for the tropical (June-November) and extra-tropical (December-May) seasons. This is to investigate if dependence between the different flooding drivers is relatively higher in one of the seasons and to assess if there are any spatial patterns to

these differences. Our third and final objective is to compare the dependence structures of different combinations of flooding drivers derived from observations to those derived from model hindcast/reanalysis data. Comparing dependence structures across different datasets is something only very few studies have addressed to date (Paprotny et al., 2020; Ganguli et al., 2020; Zscheischler et al., 2021). This last analysis step will show how well models capture dependence structures between flooding drivers and identify the pairs of drivers and locations where model results overestimate or underestimate the dependence.

The paper is structured as follows. The datasets and methods are detailed in Sect. 2 and Sect. 3, respectively. The results are presented in Sect. 4, key findings are discussed in Sect. 5, and conclusions are given in Sect. 6.



## 2 Data

We use both observational data (for objectives 1 to 3) and model hindcast data (for objective 3) for multiple locations around the CONUS coastline. The four flood generating variables considered here are storm surge (S), waves (W), river discharge

(Q), and precipitation (P); waves are characterized by the significant wave height. In the following subsections, we first describe the observational data, directly followed by the hindcast data; in the case of waves we use two different model-based data sets (we refer to them as a hindcast data set and a reanalysis dataset) due to the absence of long observational records from wave buoys. Importantly, the hindcast data that we use for the different variables were all derived with coherent forcing from the ERA5 atmospheric reanalysis (Hersbach et al., 2020), thereby avoiding inconsistencies stemming from using different

reanalysis products.

### 2.1 Storm surge

We use hourly sea level data from the National Oceanic and Atmospheric Administration (NOAA; http://tidesandcurrents.noaa.gov/) database. Following Rashid et al. (2019), we identify 35 sites (Fig. 1) with long records extending back to 1950 or earlier and where time series at individual sites are 80% or more complete. We used the R package

'rnoaa' (Chamberlain et al., 2016) to retrieve the hourly data, year-by-year starting in 1900, via the website API. Next, the hourly time series are detrended to remove the effects of sea-level rise and variability (i.e., annual mean sea level values are derived and subtracted). Following that, the Unified Tidal Analysis and Prediction UTide package in Matlab (Codiga, 2011) is used to perform a harmonic tidal analysis on a year-by-year basis to obtain tidal constituents, using the standard set of 67 harmonic constituents. The predicted astronomical tides are then subtracted from the detrended hourly sea level time series to

derive the non-tidal residual, which is used herein as the storm surge component.



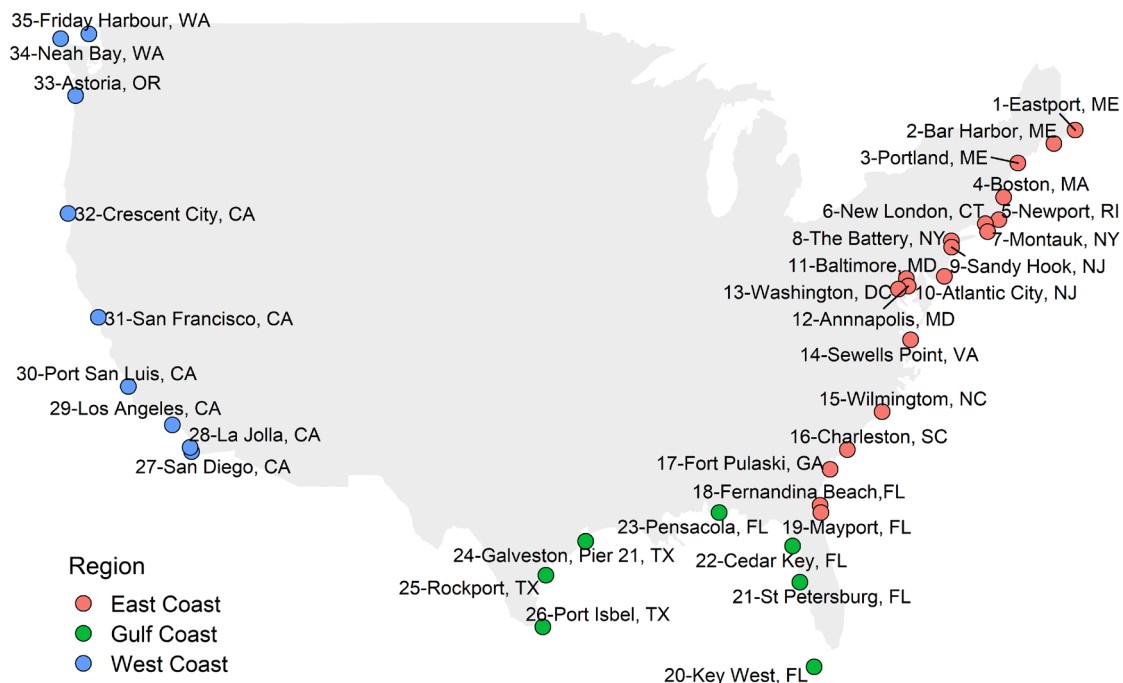

**Figure 1: Selected study sites based on tide gauge data availability and separated into east coast, Gulf coast, and west coast locations.**

Hourly model-based storm surge time-series were derived from the Coastal Dataset for the Evaluation of Climate Impact (CoDEC) (Muis et al., 2020). CoDEC was generated by forcing the third generation Global Tide and Surge Model (GTSM
v3.0), with a coastal resolution of 2.5 km globally (1.25 km in Europe), with meteorological fields from the ERA5 climate reanalysis (Hersbach et al., 2020) to simulate extreme sea levels for the period 1979 to 2017. The validation against observed sea levels demonstrated a good performance, with the annual maxima having a mean bias 50% lower than that of the previous Global Tide and Surge reanalysis dataset (GTSR) (Muis et al., 2016). We use the surge component from the model grid point that provides the maximum Kling Gupta Efficiency (KGE) (Gupta el al., 2009) from the closest 9 grid points to each tide gauge
location. The KGE metric compares observations and simulations using linear correlation, variability, and bias.

**2.2 Waves**

As outlined above we only consider the significant wave height, which is one of the most important wave parameters to represent the wave climate. In-situ observations from wave buoys are limited temporally along the US coast with lengths often restricted to 10-15 years, and hence much shorter than the time series we have available for the other flooding drivers. In
addition, the spatial coverage is sparse making it difficult to find relatively long wave records in the vicinity of the tide gauges with long records. Therefore, we use hourly hindcast wave data obtained from the US Army Corps of Engineers Wave Information Studies (USACE-WIS) (http://frf.usace.army.mil/wis/) as a substitute for observational data. USACE-WIS is a



regional product that has been widely applied for engineering purposes and extensively validated against wave buoy observations; it provided wave information for the US coastlines for over 30 years with continuous development of hindcasts

and evaluation of model results and technology. Models are forced with winds generated from the National Center for Atmospheric Research NCEP-NCAR Reanalysis 1 (spatial resolution = 0.5°×0.5° and temporal resolution = 6 hours). The Atlantic, Pacific, and Gulf of Mexico are each modelled independently for best results with a coastal grid resolution of 5 mins and temporal coverage from 1980-2014. We select the WIS grid points that are closest to the tide gauge locations. We do not pair tide gauges that are located further upstream in estuaries with wave data, as these locations are often sheltered for wave

action, which leads to 31 sites where co-located data are identified and used for the analysis.

We compare the WIS data against wave time-series extracted from the ERA5 reanalysis (spatial resolution = 0.5°×0.5° and temporal resolution = 1 hour) (Hersbach et al., 2020) based on the WAM model. For our analysis we use the grid points closest to the WIS gird points selected before.

## 2.3 River discharge

We obtained observed river discharge time series from the United States Geological Survey (USGS) National Water Information System (NWIS) (https://waterdata.usgs.gov/nwis). USGS-NWIS provides nationwide water flow (and quality) information in streams and lakes. The R package 'dataRetrieval' (De Cicco et al., 2018) was used for retrieving data from desired locations close to the 35 tide gauge sites identified before. The selected stream gauges were chosen to satisfy the following criteria: (1) a minimum catchment area of 1000 km2, (2) a maximum Euclidean distance to the matching tide gauge

< 500 km, (3) a river basin outlet within a maximum distance of 55 km (0.5 degree) from the tide gauge (Ward et al., 2018), and (4) to lead to overlapping records with the tide gauges of 20 years or more. Based on these rules we identify 23 sites where tide gauge data can be paired with discharge data.

Modelled river discharge time series were extracted from the Global Flood Awareness System (GloFAS)-ERA5 reanalysis (Harrigan et al., 2020). This is a global gridded reanalysis dataset (excluding Antarctica), with a horizontal resolution of

0.1°×0.1° at a daily time step over 40 years starting 1979. The GloFAS-ERA5 river discharge reanalysis was produced by coupling the land surface model runoff component of the ECMWF ERA5 global reanalysis with the LISFLOOD hydrological and channel routing model. LISFLOOD allows the lateral connectivity of ERA5 runoff grid cells routed through the river channel to produce river discharge. ERA5 runoff is produced from the HTESSEL land surface model (Hydrology Tiled ECMWF Scheme for Surface Exchanges over Land) with an advanced land data assimilation system to assimilate conventional

in-situ and satellite observations for land surface variables. We again identify the 9 gird points closest to the stream gauges selected before and retain the ones with the highest KGE statistic.

## 2.4 Precipitation

We use precipitation observations from the Global Historical Climatology Network Daily (GHCN-D) hosted by NOAA's National Centers for Environmental Information (NOAA-NCEI) (https://www.ncdc.noaa.gov/ghcnd-data-access). For data



retrieval we used the R package 'rnoaa' (Chamberlain et al., 2016). GHCN-D contains precipitation and other climate data from more than 100,000 stations worldwide covering periods ranging from 1 year to over 175 years. We consider the accumulated daily precipitation depth from rain gauges that are located closest to the selected tide gauges with at least 20 years of overlapping data; in 31 instances the closest precipitation gauges providing long records are found within a 30 km radius around the respective tide gauges, for the other 4 sites the distance is larger but always smaller than 60 km.

Model-based precipitation time-series were extracted from the ERA5 reanalysis which is based on the Integrated Forecasting System (IFS) cycle 41r2. The ERA5 reanalysis replaces the ERA-Interim reanalysis with a significantly enhanced horizontal resolution of 31 km (~ 0.25°×0.25°), compared to 80 km for ERA-Interim. In addition, biases are strongly reduced in ERA5 compared to ERA-Interim precipitation data. The ERA5 hourly dataset spans 1979 onwards and we used that to derive accumulated daily precipitation. Similar to the other drivers we selected the 9 grid points closest to the precipitation gauges

and selected the ones with the highest KGE statistic.

### 2.5 Final study sites

Following the procedure outlined above leads to a dataset that comprises information on storm surges, significant wave height, precipitation, and river discharge derived from observations and model hindcasts for 35 sites around the US coast (see Fig. 1); the overlapping record lengths between the various data pairs considered in the compound flooding potential analysis range

from 20 years to 100 years (mean = 47 years, median = 35 years) (Fig. 2).

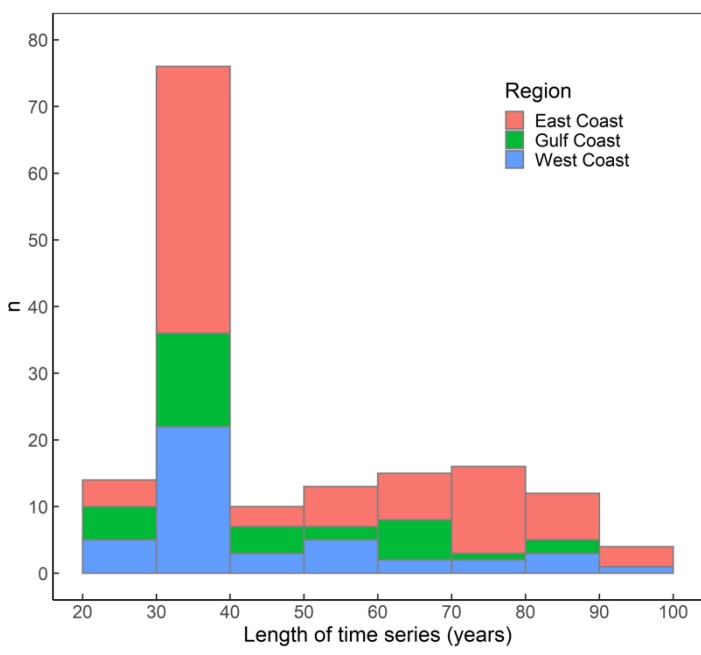



**Figure 2: Histogram showing the length of overlapping time series in years (on the x-axis) and the corresponding frequency n (on the y-axis) for the observational data used in the analysis (lengths of model data sets are outlined in the text and the same for all locations for a given variable); note that we analyse 35 locations and a maximum of 6 driver combinations (from 4 drivers) at each location, all of which were used to derive the histogram. Bars are separated into East coast, Gulf coast, and West coast locations.**

## 3 Methods

Our analysis is performed in three stages each corresponding to one of the three objectives outlined in Sect. 1. These are described in turn below.

### 3.1 Dependence Analysis

Our first objective is to characterize and map the dependence between different flood driver combinations at the 35 sites around the CONUS coastline and identify spatial patterns. First, we derive daily data for all four variables. For storm surge and significant wave height these are the maximum hourly values that occurred during a given day, for precipitation we use the accumulated daily precipitation depth, and for discharge we use the daily mean. From these daily time series, we further identify extreme events using the annual block maxima method. This avoids having to select appropriate thresholds for all sites and variable pairs as would be required when using the peaks-over-threshold method; both approaches were contrasted in a comprehensive sensitivity analysis by Camus et al. (2021) who found comparable results.

We use six combinations of variables and for each we apply two-way conditional sampling similar to previous studies (Wahl et al., 2015; Ward et al., 2018; Couasnon et al., 2020; Jane et al., 2020; Santos et al., 2021; Camus et al., 2021), where at least one variable is extreme. In particular, we use annual maxima of the first (conditioning) variable and the corresponding maximum value of the other (conditioned) variable within a time window that we vary between 0 to ±10 days; the relatively long lag-times are chosen as we do not correct for the travel time of river flow from where it is measured/modelled and the tide gauge further downstream. The six variable pairs (and 12 combinations from the two-way sampling) are the following; the variable that is listed first if the conditioning variable (e.g., S_Q means that annual maxima S is paired with (near-) coincident Q, whereas Q_S means that annual maxima Q is paired with (near-)coincident S):

Surge and discharge (S_Q & Q_S);

Surge and Precipitation (S_P & P_S);

Surge and Waves (S_W & W_S);

Discharge and precipitation (Q_P & P_Q);

Discharge and waves (Q_W & W_Q); and

Precipitation and waves (P_W & W_P).

We assess dependence using Kendall's rank correlation coefficient ($\tau$) (Kendall, 1938) which, in contrast to Pearson's linear correlation coefficient, can also capture non-linear dependence between the variable pairs and was used in previous studies to assess dependence (e.g., Wahl et al., 2015; Ward et al., 2018; Hendry et al., 2019; Marcos et al., 2019). Another option would be to use Spearman's rank correlation coefficient ($\rho$) which measures the strength of monotonic dependence between bivariate



variables (e.g., Couasnon et al., 2020). Camus et al. (2021) compared both measures and found that Spearman's rank correlation coefficient was typically higher than Kendall's rank correlation coefficient. However, both showed the same spatial characteristics when applied to many locations along the European coastline. Significance is assessed here at $\alpha = 0.05$ (i.e., 95% confidence level).

In addition to using Kendall's $\tau$ in association with the two-way sampling approach, we also assess extremal dependence using

tail dependence coefficients. In this method, extremal (or tail) dependence falls into two categories: (1) asymptotic tail dependence or (2) asymptotic tail independence (Ledford and Tawn, 1997). If (A, B) are a pair of variables with cumulative distribution functions $(F_a, F_b)$ transformed to unit scale (0,1), (U $= F_a$(A), V$=F_b$(B)). Then (A, B) are asymptotically tail dependent if

$$\chi = \lim_{q \to 1} P\ (F_a A > q | F_b B > q)\ \epsilon\ (0,1]$$

and asymptotically tail independent if $\chi = 0$. The coefficient $\chi$ represents the probability of one variable being extreme (exceeding a threshold q) given that the other variable is extreme (exceeding the same threshold q). We choose q= 0.9 (90[th] percentile) following previous studies (e.g., Vignotto et al., 2021). We estimate $\chi$ using the function 'taildep' from the R package extRemes (Gilleland and Katz, 2016). To estimate whether the calculated $\chi$ values are significant, a bootstrap method following Svensson and Jones (2002) is implemented. Data is bootstrapped randomly by shuffling the temporal order of one

variable (using blocks of 1-year length) to break up the dependence structure while preserving seasonality. This is repeated 1,000 times and if less than 5% of the bootstrapped estimates are greater than $\chi$ calculated from the original records, then $\chi$ is considered significant.

Note that in the extremal dependence method, the bivariate daily time series are used and extremal dependence is assessed for all values exceeding the predefined threshold without any de-clustering to obtain independent events. In contrast, in the rank

correlation analysis, $\tau$ is calculated for a time series of annual maxima of a variable and maximum value of another variable within a time window. These differences in the experiment design will inevitably lead to differences in the results, but both metrics provide important insight into the existence (or non-existence) of dependence and its structure.

### 3.2 Seasonal Dependence Analysis

Our second objective is to perform a seasonal dependence analysis where we analyse data from the tropical cyclone season

(June-November) separately from the extra-tropical season (December-May). Data from each season were studied separately and compared to investigate if dependence varies between them. The analysis is performed in the same way as outlined in Sect. 3.1, i.e., for the rank correlation analysis instead of using annual maxima we use seasonal maxima of the conditioning variables and match those with near-coincident values of the conditioned variables. The tail dependence analysis is conducted separately for both seasons using daily data corresponding to each season.

To assess the significance of the difference in dependence and tail dependence between seasons, confidence intervals were calculated for each statistic (Kendall's $\tau$ and tail dependence $\chi$) using a bootstrapping method similar to Svensson and Jones



(2004) and Wahl et al. (2015). This is done by generating many new datasets from the existing dataset through resampling (sampling with replacement). Unlike the bootstrapping method explained in Sect. 3.1 where significance was assessed based on independence, here we sample (with replacement) both pairs at the same time: for Kendall's $\tau$ we draw a bivariate

observation (in a season-year) while for tail dependence $\chi$ we draw a bivariate block (of length 1 season-year) at a time. To ensure that each season-year is sampled equally often, a balanced resampling approach was implemented, which avoids bias from certain years being sampled more often than others. For each season, a two-column matrix with N*B rows is created, where N is the length of the overlapping data of a certain pair of flooding drivers and B is set to 1,000. The resulting matrix is then shuffled while keeping the bivariate pairs intact and afterwards sliced into B slices of length N. For each of the B matrices

of length N the desired statistics (Kendall's $\tau$ or tail dependence $\chi$) are calculated and the 95% confidence intervals are estimated (2.5% and 97.5% quantiles). The confidence intervals derived for each season are compared and if they do not overlap then we consider the difference in dependence (expressed as $\tau$ or $\chi$) to be significant.

### 3.3 Observation-based vs Model-based Dependence Structure

Our third and final objective is to compare the dependence structures derived from observation-based data and model-based

data. We perform this part of the analysis only for the extremal (tail) dependence $\chi$. This is because the two-way sampling approach uses the annual (or seasonal) maxima values of the different variables, and those are often not well captured by model hindcasts leading to a higher sensitivity in the results as opposed to the tail dependence which uses the full daily time series, and here a threshold of q = 0.9. We calculate the extremal (tail) dependence $\chi$ (at q=0.9) from observation and hindcast data for periods where data from both sources are available (Paprotny et al., 2020). We apply the Kullback-Leibler (KL) divergence

to assess significance in the difference in tail dependence derived from the two types of data. The method based on KL divergence has been introduced by Zscheischler et al. (2021) to assess if the dependence structure between wind and precipitation extremes was different across different datasets in a study location in Europe. The method builds on earlier work of Naveau et al (2014) for comparing univariate datasets and extends it to bivariate datasets. Vignotto et al. (2021) also used the KL divergence for clustering bivariate dependencies of compound precipitation and wind extremes over Great Britain and

Ireland.

We provide a brief description of the methodology (see Zscheischler et al. (2021), Vignotto et al. (2021), and references therein for more details). For two bivariate distributions $X^{(1)} = \left(X_1^{(1)}, X_2^{(1)}\right)$ and $X^{(2)} = \left(X_1^{(2)}, X_2^{(2)}\right)$, corresponding to bivariate distributions from observation-based and model-based data, the divergence is only defined in the tail of the distributions after normalizing the marginal distributions to standard Pareto distributions. A risk function (r: $R^2 \longrightarrow R$) calculated on the Pareto

scale is used to define extremal regions on each of the bivariate distributions. From the risk functions introduced in Zscheischler et al. (2021) we choose the 'minimum' corresponding to r(**x**) = min ($x_1, x_2$), with **x** = ($x_1, x_2$) as it covers both asymptotically dependent and independent data. This results in two univariate variables: $R^{(1)} = r(X^{(1)})$ and $R^{(2)} = r(X^{(2)})$. We consider points as extreme when the variable $R^{(j)}$ exceeds a given high quantile threshold $q_u^{(j)}$ corresponding to an


exceedance probability $u \in (0,1)$, $j = 1,2$. Varying the threshold $q_u^{(j)}$ changes the extremal region of interest (we used u = 0.9
to be consistent with the tail dependence threshold we employed). Applying the minimum risk function for each of the two

bivariate distributions, the extreme points are contained in the set $\{R^{(j)} > q_u^{(j)}\}$, $j = 1,2$. This set is then divided into a fixed

number of disjoint sets $A_1^{(j)}, \dots, A_W^{(j)}$. For the minimum risk function the data is partitioned into W = 3 sets where one contains

the co-occurring extremes and the other two contain data when only one variable is extreme.

For the two random samples $(X_1^1, \dots, X_n^1)$ and $(X_1^2, \dots, X_n^2)$ from the two distributions $X^{(1)}$ and $X^{(2)}$, the empirical proportions
of data points in each of the previously mentioned sets $A_w^{(j)}$ are computed as:

$$\hat{p}_w^{(j)} = \frac{\#\{i : X_i^{(j)} \in A_w^{(j)}\}}{\#\{i : r\left(X_i^{(j)}\right) > q_u^{(j)}\}}, \qquad w = 1, \dots, W; \; j = 1,2; \; i = 1, \dots, n.$$

The difference between the extremal behaviours of the two distributions can be measured as the KL divergence between the
two multinomial distributions defined through the previous empirical proportions as follows

$$d_{12} = D\left(X_1^{(1)}, X_2^{(1)}\right) = \frac{1}{2} \sum_{w=1}^{W} \left( \left(\hat{p}_w^{(1)} - \hat{p}_w^{(2)}\right) \log\left(\frac{\hat{p}_w^{(1)}}{\hat{p}_w^{(2)}}\right) \right)$$

The divergence $d_{12}$ is a natural way to contrast the differences between extremal dependence structures for asymptotically

dependent and independent data. Also, this divergence is symmetric and does not require additional model assumptions as it

is a non-parametric statistic. The statistic $d_{12}$ follows a $\chi^2(W - 1)$ distribution in the limit as the sample size approaches $\infty$

under suitable assumptions allowing us to estimate whether it is significantly different from zero.

We repeat the analysis after splitting the dataset into tropical and extra-tropical seasons to investigate if models' performance
is better in one season compared to the other.

## 4 Results

### 4.1 Overall Dependence Analysis

This section describes the results for the first objective, relating to the bivariate dependence analysis between the four drivers.

First, we show the results from Kendall's rank correlation analysis applied to the two-way samples derived with the annual
maxima method, and then we show results from extremal (tail) dependence (that will be referred to as tail dependence

hereafter).

In Fig. 3, the dependence based on Kendall's $\tau$ is shown between all combinations of drivers at the 35 study sites around the

CONUS coastline. Sites where one driver was not available or where the number of overlapping years between bivariate

drivers was less than 20 are blank and insignificant dependence (at $\alpha = 0.05$) is shown as an asterisk (*). For surge and





discharge, out of 23 sites analysed, more sites show significant correlation for Q_S (14 sites) than S_Q (11 sites). Along the coasts of Florida and US southeast higher values of τ are found for S_Q than Q_S. In contrast, along the coasts in the western Gulf of Mexico and US southwest, the values of Q_S are higher than for S_Q. For surge and precipitation, from the 35 sites analysed, more sites show significant dependence in S_P (24) compared to P_S (16). Along the East coast, more sites possess significant dependence in S_P (14 compared to 9 in P_S) and at the same time the dependence values for S_P are higher than

P_S values. Interestingly, along the Gulf and West coasts, although more sites have significant dependence in S_P (10, compared to 7 in P_S), the strength of the dependence is higher for P_S in most cases (6 out of the 7 sites); this is in agreement with results from Wahl et al. (2015). For surge and waves, out of 31 sites analysed, we find more sites with significant dependence in the case of S_W (25) compared to W_S (16), especially along the East coast. For the East and Gulf coasts, the strength of dependence is also higher for S_W compared to W_S, which is reversed on the West coast. For discharge and

precipitation, out of 23 sites analysed, more sites show significant dependence in the case of Q_P (17) compared to P_Q (13), which is again most pronounced on the East coast. The strength of dependence of Q_P and P_Q in the Gulf and West coasts is higher than that for the East Coast. In most sites, the strength of dependence is higher for Q_P than P_Q. For waves and discharge, out of 17 sites analysed, only three show significant dependence in both cases. For W_Q all three are located in Florida and show relative high dependence strength. Lastly, for wave and precipitation, out of 31 analysed sites, there is

approximately an equal number of sites showing significant dependence for W_P (12) and P_W (11). However, the strength of dependence is overall higher for W_P compared to P_W at most sites (4 of 5) where both are significant.

In general, our results indicate from a geographic perspective that dependence, when assessed through Kendall's τ, is higher between most drivers along the Gulf, southeast, and southwest coasts compared to northeast and northwest coasts. From a flooding driver perspective, the highest dependence is found between surges and waves which are both oceanographic drivers,

followed by surge and precipitation, surge and discharge, waves and precipitation, and waves and discharge.





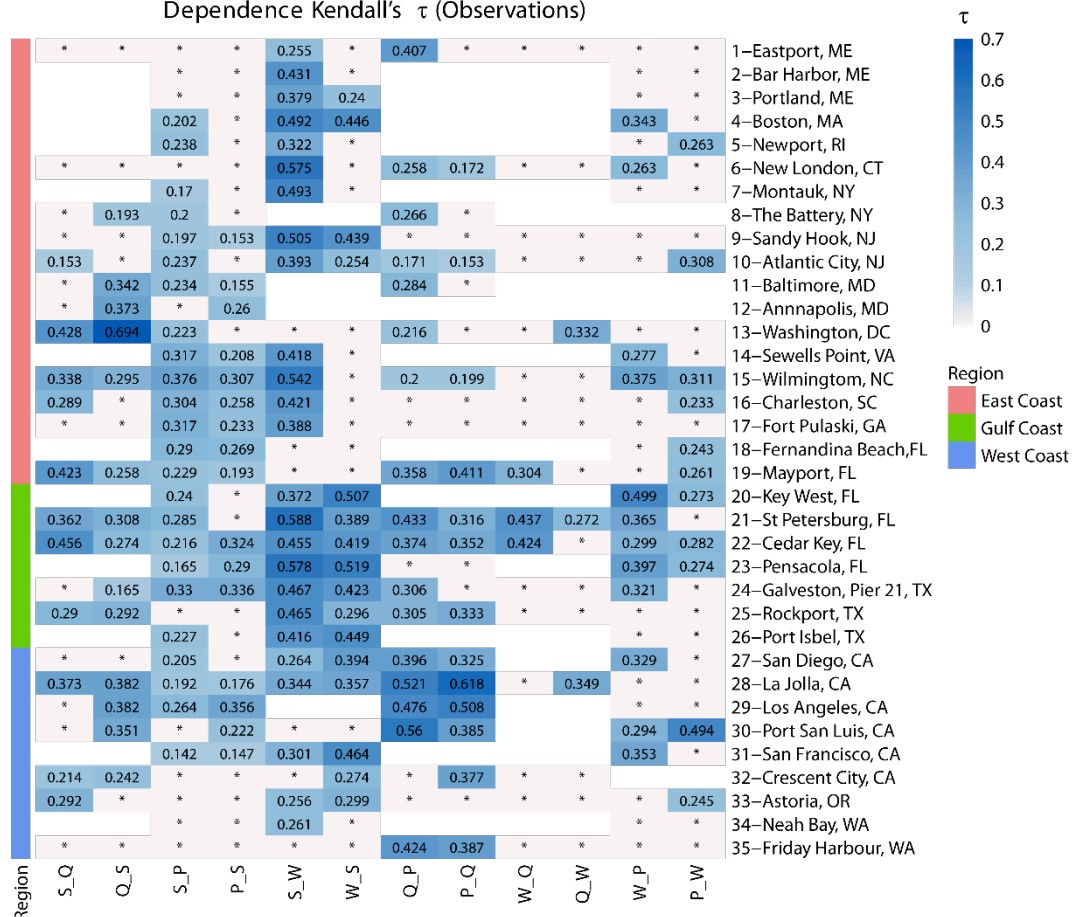

**Figure 3: Dependence between different pairs of flooding drivers based on Kendall's τ and two-way sampling using annual maxima. Sites are grouped into East, Gulf, and West coast locations (see colours on the left and legend). The blue colour bar denotes dependence strength, blank squares indicate that data for the particular pair didn't exist or that the number of overlapping years was less than 20 and squares with \* indicate that correlation is not significant.**

The results from the tail dependence analysis using $\chi$ calculated at q =0.9 are shown in Fig. 4. Recall that for the calculation of $\chi$ we consider the full daily time series of all variables; hence we only obtain results for one case as opposed to the results presented above which are based on the two-way sampling procedure. The results for the tail dependence analysis indicate that there are more sites with significant tail dependence compared to the two-way sampling analysis with Kendall's τ. Geographically, we find more places with significant tail dependence in the northwest coast for the pairs S_P, W_Q, and W_P whereas the rank correlation analysis using Kendall's τ pointed to insignificant correlation between the same pairs. Nevertheless, in terms of the strength of dependence between different variable pairs the order found with Kendall's τ persists.





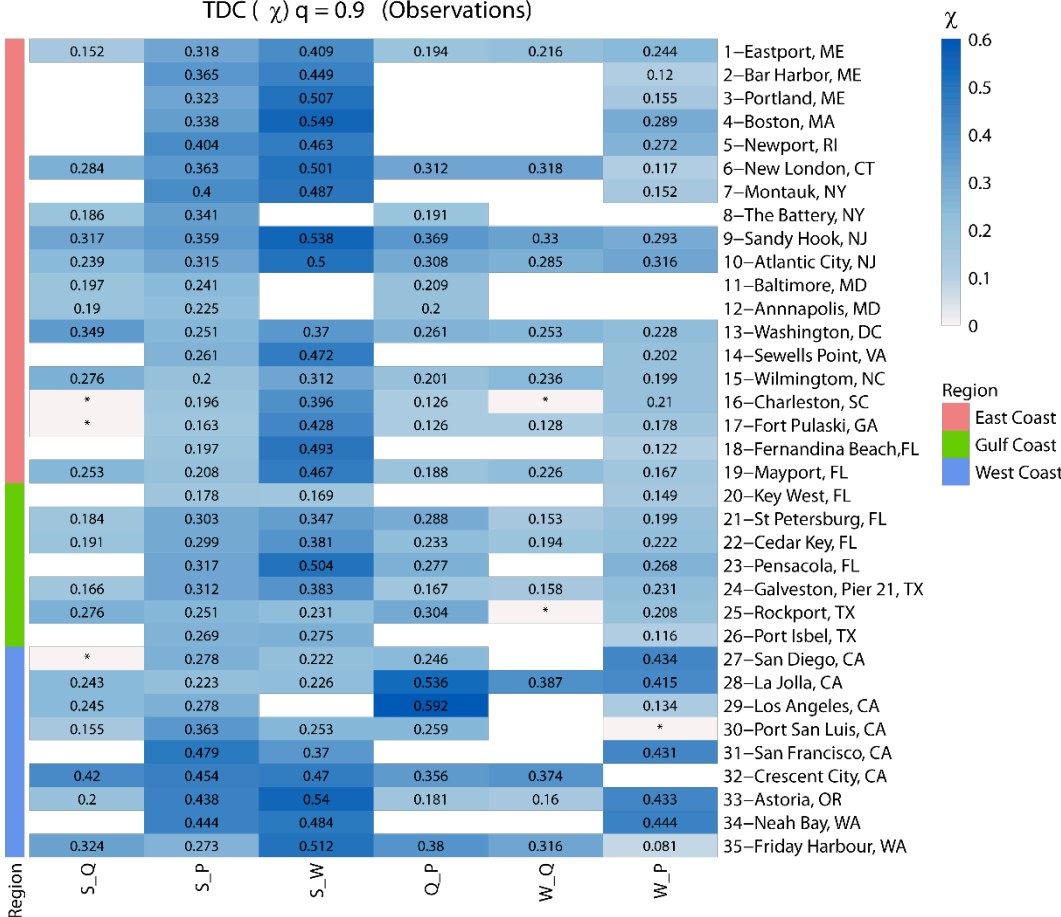

**Figure 4: Tail dependence χ between different pairs of flooding drivers for a threshold of q = 0.9. Sites are grouped into East, Gulf, and West coast locations (see colours on the left and legend). The blue colour bar denotes tail dependence strength, blank squares indicate that data for the particular pair did not exist or that the number of overlapping years was less than 20, and squares with \* indicate that dependence is not significant.**

### 4.2 Seasonal Dependence Analysis

This section describes the results for the second objective, relating to the seasonal dependence analysis between the four drivers. Here we analyse data from the tropical cyclone season (June-November) separately from the extra-tropical season (December-May) and compare results. First, we show the results from Kendall's rank correlation analysis applied to the two-way samples derived with the seasonal maxima method, and then we show results from analysing tail dependence.

The comparison between dependence using Kendall's τ in the tropical season (plotted on the x-axis) and extra-tropical season (plotted on the y-axis) is shown in Fig. 5. Note, that for the scatter plot we did not distinguish between the two cases, i.e., the pair Q_S that is shown as filled circles includes the results for both Q_S and S_Q cases. Overall, the values are dispersed widely from the diagonal 1:1 line (R = 0.42 when all Q_S and S_Q samples) indicating the existence of differences in the dependence across the two seasons where different types of storms are dominant. The deviation from the diagonal is more



pronounced in the lower left where the majority of markers are located, whereas markers tend to be closer to the diagonal for sites/pairs where the dependence is generally higher. In the majority of cases, the dependence values tend to be stronger in the

tropical season (as indicated by the dashed lines in Fig. 5 representing linear regression fits to the data subsets for different regions). This is particularly notable for the Gulf coast (shown in green), where the majority of markers fall below the diagonal indicating stronger dependence in the tropical season. This tendency also exists for the East coast sites, but much less pronounced, whereas for the West coast sites the markers are scattered more symmetrically around the diagonal.

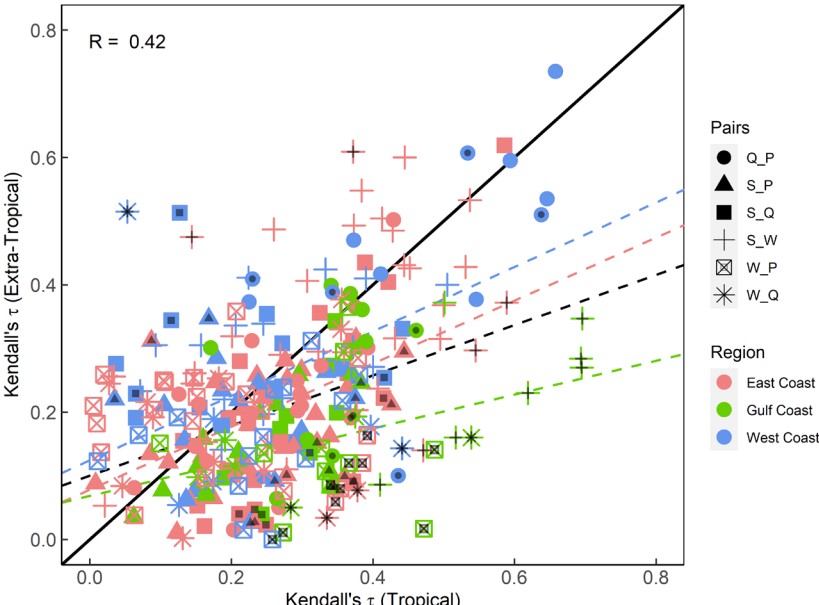

**Figure 5: Scatter plot comparing dependence derived with Kendall's τ and two-way sampling using seasonal maxima approach for tropical and extra-tropical seasons. Colours denote the location (separated into East, Gulf, and West coast) and markers represent the different variable pairs. Black dots on markers indicate significant difference in dependence between seasons. Dashed lines show linear regression fits corresponding to all data points (black) and for different subsets according to locations (coloured as outlined in the legend).**

To better discern spatial patterns where differences in the seasonal dependences for certain variable pairs are larger, Fig. 6 shows the same results as in Fig. 5 but separately for each of the 12 variable pairs (considering both cases of the two-way sampling) and all individual sites. For the pairs S_Q, Q_S, S_P, and P_S higher values of τ are found along the Gulf and East coasts for the tropical season, while higher values are found for the extra-tropical season on the West coast. For surge and waves, higher dependence is found for both pairs (S_W and W_S) in the Gulf of Mexico during the tropical season and the

difference is significant. S_W is higher during the tropical season on the East coast, and lower on the West coast. In contrast, W_S is lower during the tropical season on the East, and higher on the West coast. For the rest of the pairs (Q_P, P_Q, W_Q, Q_W, W_P, P_W) the dependence is overall higher during the tropical season compared to the extra-tropical season in the Gulf of Mexico, whereas mixed patterns are found along the East and West coasts.





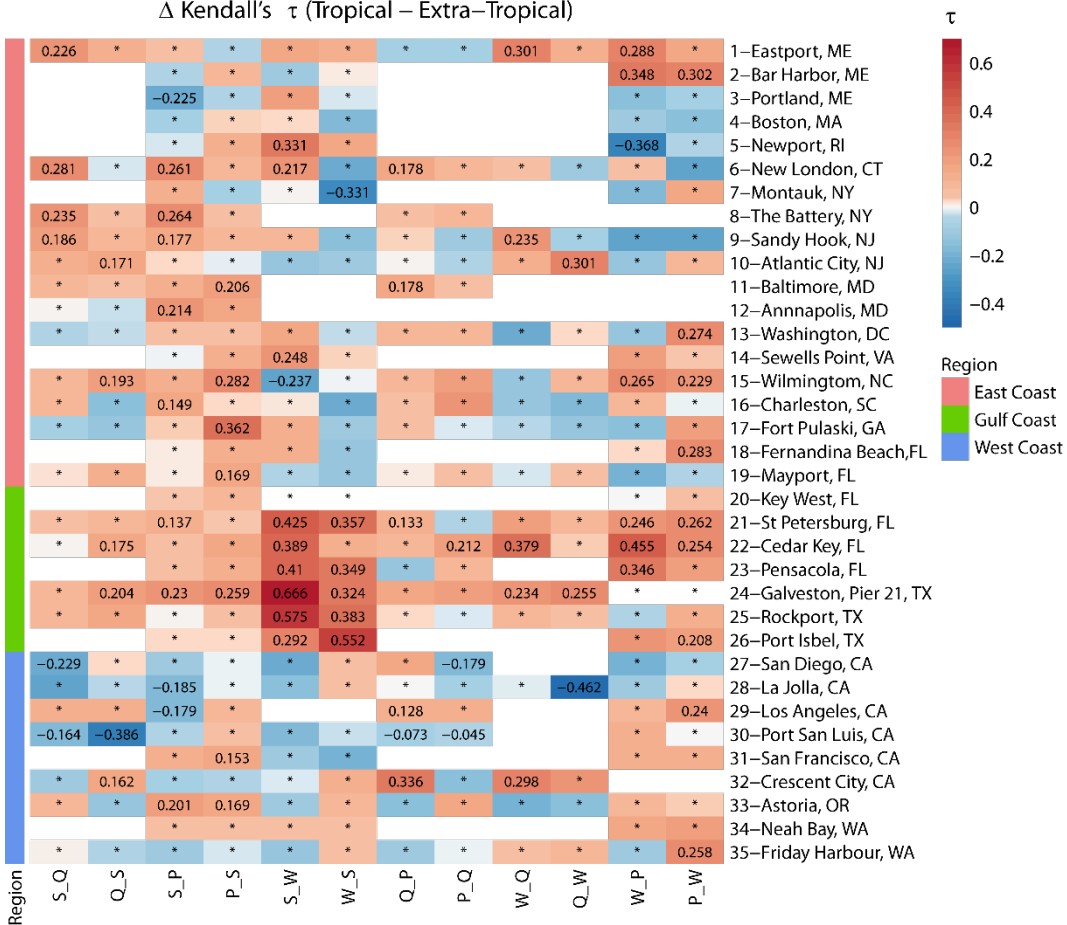

**Figure 6: Heat map showing differences in Kendall's τ derived from two-way sampling using seasonal maxima approach for tropical and extra-tropical seasons. Sites are grouped into East, Gulf, and West coast locations (see colours on the left and legend). The colour bar denotes the difference between τ in the tropical versus extra-tropical season, where red colour denotes higher dependence in the tropical season, and blue colour denotes higher dependence in the extra-tropical season. Squares with * indicate that difference in dependence across seasons is not significant. Blank squares indicate that data for the particular pair did not exist or that the number of overlapping years was less than 20.**

Similar to the overall dependence analysis, we also assess differences in seasonal tail dependence using χ. The results are shown in Fig. 7 and Fig. 8. Interestingly, the results point to different patterns as we found from the dependence analysis based on Kendall's τ. In Fig. 7, markers for different pairs are scattered more closely around the diagonal (1:1 line) with a Pearson correlation coefficient of R=0.75 indicating more similarity across seasons. For the West coast, many markers (especially S_P, S_Q, and Q_P) above the diagonal indicate stronger tail dependence in the extra-tropical season while for the East and Gulf coasts results are more symmetric.

Discrepancies found in results when comparing between seasons using tail dependence χ and Kendall's τ are mainly due to the sample from which each statistic is calculated. For tail dependence, all bivariate daily values exceeding a certain threshold (q=0.9) are used while for calculating Kendall's τ two-way sampling using block (seasonal or annual) maxima is used. Two-





way sampling makes members of the samples independent and identically distributed (1 value is picked per block) while
excesses above a certain threshold that are used for χ are not declustered.

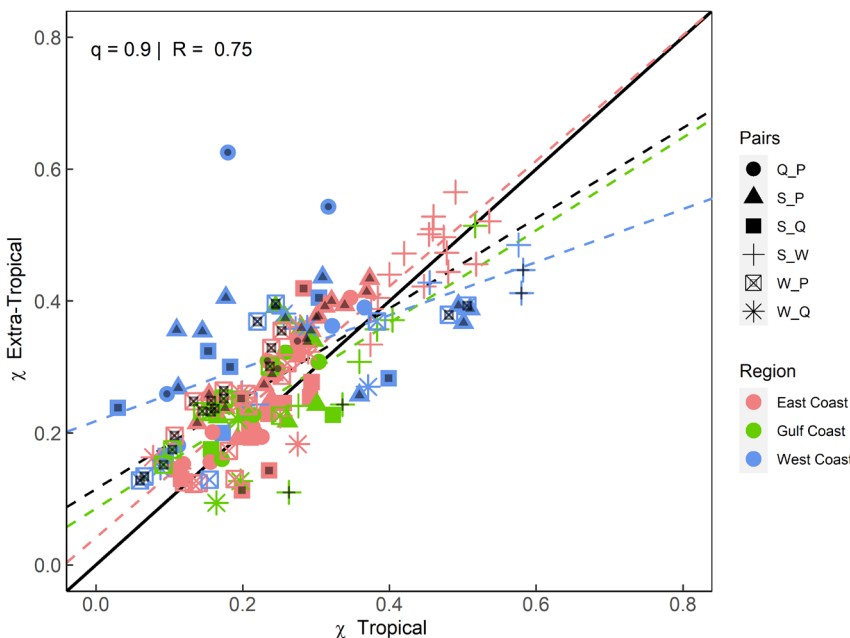

**Figure 7: Scatter plot comparing tail dependence (for q=0.9) derived for tropical and extra-tropical seasons using daily time series of both variables. Colours denote the location (separated into East, Gulf, and West coast) and markers represent the different**
**variable pairs. Black dots on markers indicate significant difference in tail dependence between seasons. Dashed lines show linear regression fits corresponding to all data points (black) and for different subsets according to locations (coloured as outlined in the legend).**





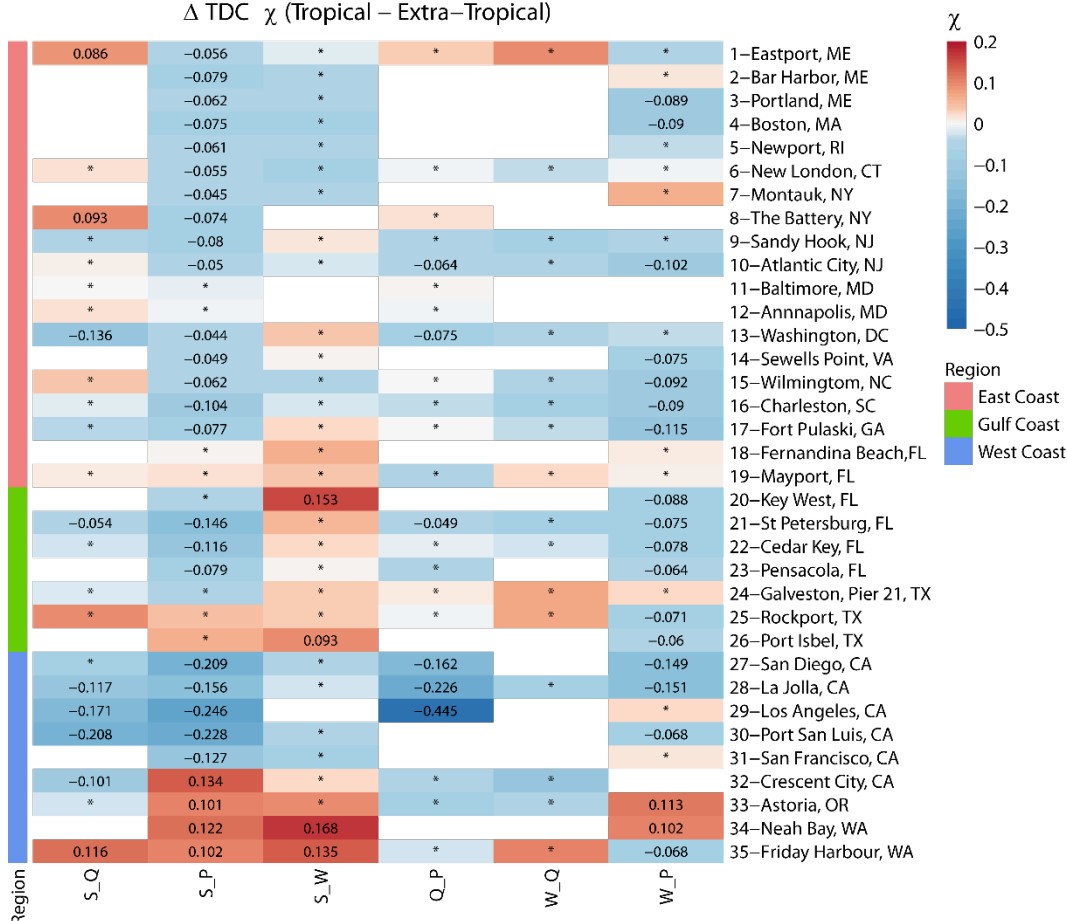

**Figure 8: Heat map showing differences in tail dependence (for q=0.9) derived for tropical and extra-tropical seasons using daily time series of both variables. Sites are grouped into East, Gulf, and West coast locations (see colours on the left and legend). The colour bar denotes the difference between χ in the tropical versus extra-tropical season, where red colour denotes higher dependence in the tropical season and blue colour denotes higher dependence in the extra-tropical season. Squares with \* indicate that difference in dependence across seasons is not significant. Blank squares indicate that data for the particular pair did not exist or that the number of overlapping years was less than 20.**

## 4.3 Observation-based vs Model-based Dependence Structure

This section describes the results for the third and final objective, relating to the comparison between the dependence structures when using model-based versus observation-based data. We perform this part of the analysis only for the tail dependence χ. This is because the two-way sampling approach uses the annual (or seasonal) maxima values of the different variables, and those are often not well captured by model hindcasts leading to a higher sensitivity in the results as opposed to the tail dependence which uses the full daily time series, and here a threshold of q = 0.9.

We start by comparing results for the full (annual) data sets and these are shown in Fig. 9 and Fig. 10. We find that in general the models perform well in capturing tail dependence (Pearson correlation R =0.75). Using KL divergence provides





complementary information on whether tail dependence structures calculated using models are significantly different from
those derived with observations. Pairs where the difference is significant are highlighted with black dots in Fig. 9. Figure 10
shows the same results but in a way that allows us to discern the spatial patterns. Both figures show that models tend to
overestimate W_P dependence in most of the analysed sites. S_Q is underestimated by models on the East and West coasts
but well captured among sites analysed in the Gulf. S_P is well captured along Gulf and southeast coasts but overestimated in
the northeast and northwest and underestimated in the southwest. S_W is overestimated in some sites in the Gulf and southeast
coast. For the rest of the pairs there is mixed behaviour with no clear spatial pattern.

We note that in some cases the difference in the tail dependence is small (i.e., markers lying on or close to the 1:1 line in Fig.
9 or showing light colour in Fig. 10) but still significantly different according to the KL divergence. This is because two
bivariate distributions with equal (or very similar) tail dependence coefficients may still vary in their dependence structure and
this cannot be assessed by just calculating the difference in $\chi$. The reason is that $\chi$ only focuses on the diagonal (Zscheischler
425 et al., 2021), whereas KL divergence partitions the extremal space defined by the risk function (above the selected threshold)
into a number of sets and thus better captures the dependence structure.

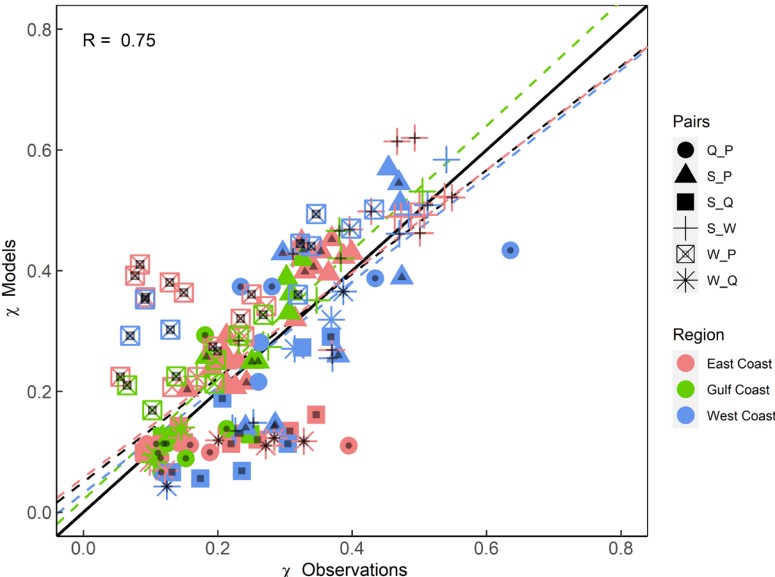

**Figure 9: Scatter plot comparing extremal (tail) dependence (for q=0.9) derived using observations (x-axis) and models (y-axis) using daily time series of both variables. Colours denote the location (separated into East, Gulf, and West coast) and markers represent**
430 **the different variable pairs. Black dots on markers indicate significant difference in tail dependence structure between observations and models. Dashed lines show linear regression fits corresponding to all data points (black) and for different subsets according to locations (coloured as outlined in the legend).**





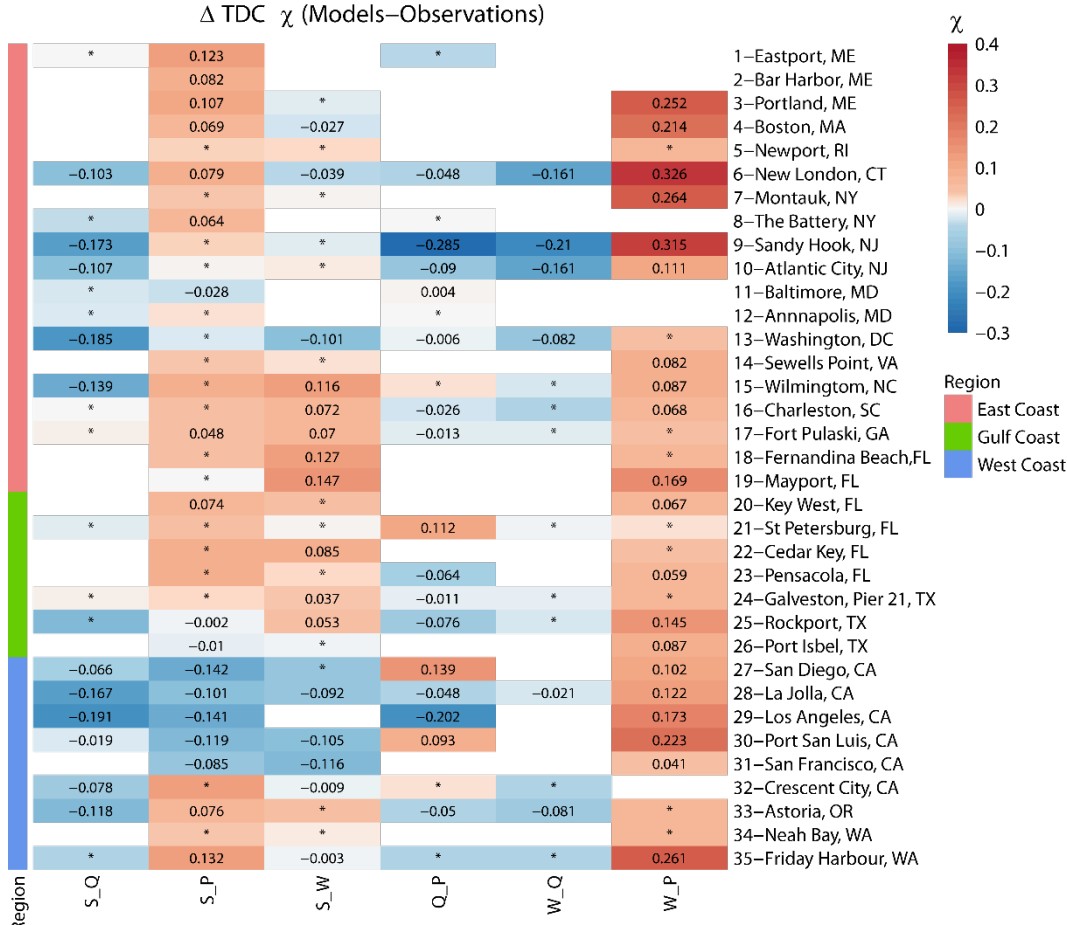

**Figure 10: Scatter plot comparing dependence derived with Kendall's τ and two-way sampling using seasonal maxima approach for tropical and extra-tropical seasons. Colours denote the location (separated into East, Gulf, and West coast) and markers represent the different variable pairs. Black dots on markers indicate significant difference in dependence between seasons. Dashed lines show linear regression fits corresponding to all data points (black) and for different subsets according to locations (coloured as outlined in the legend).**

We repeat the analysis again for the tropical and extra-tropical seasons to assess whether models perform better in one of them when tail dependence is compared to observations. Figure 11 shows that overall models perform better during the tropical season (Pearson correlation R= 0.77) in comparison with the extra-tropical season (Pearson correlation R= 0.7). Especially for higher values of χ points are more aligned with the 1:1 line for the tropical season, with a tendency of model overestimation (markers above the diagonal) for the pair S_W for several sites on the East coast. For the extra-tropical season, and for higher values of χ models tend to overestimate S_P at several sites across all coasts, with no clear pattern. Figure 12 shows that models overestimate the tail dependence between W_P everywhere during both seasons and also overestimate S_P during the extra-tropical season in the Gulf. Tail dependence between Q_P is overestimated at several sites on the West coast (in California) during the tropical season but underestimated during the extra-tropical season.



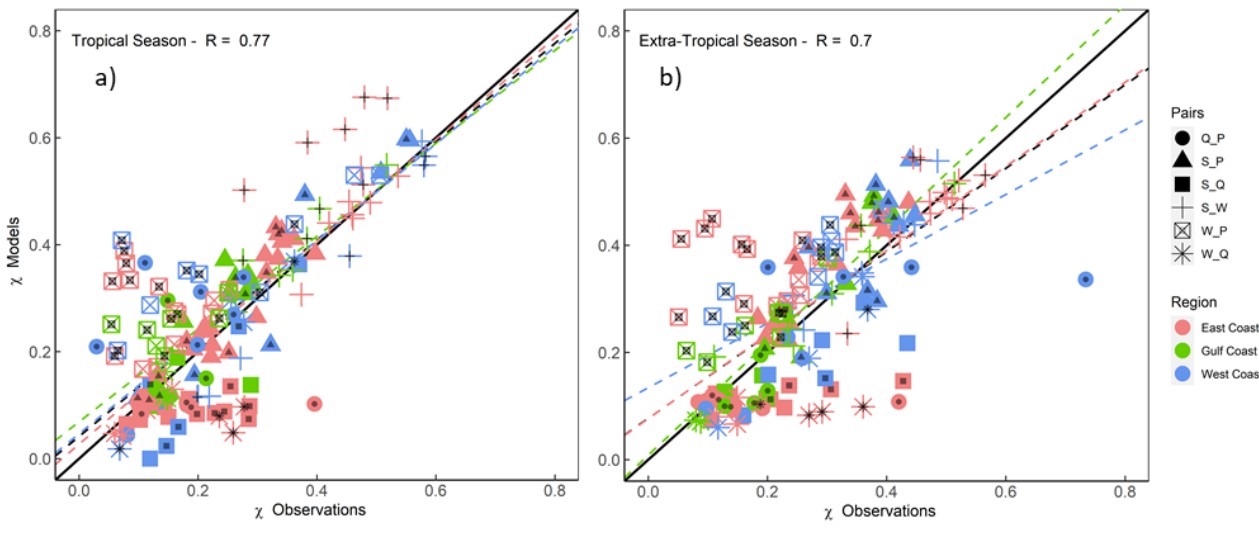

**Figure 11: Scatter plot comparing dependence derived with Kendall's τ and two-way sampling using seasonal maxima approach for tropical and extra-tropical seasons. Colours denote the location (separated into East, Gulf, and West coast) and markers represent the different variable pairs. Black dots on markers indicate significant difference in dependence between seasons. Dashed lines show linear regression fits corresponding to all data points (black) and for different subsets according to locations (coloured as outlined in the legend).**



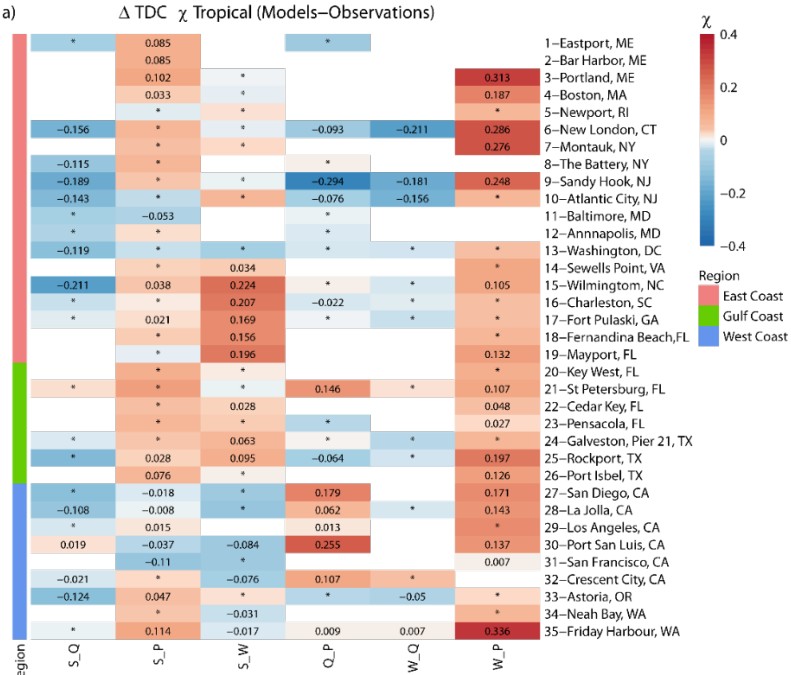

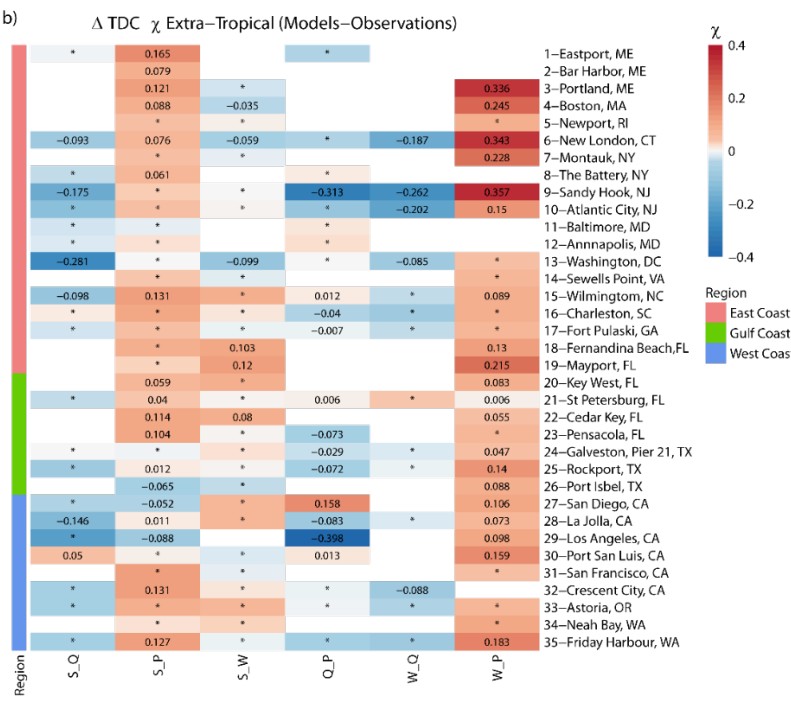

**Figure 12: Scatter plot comparing dependence derived with Kendall's τ and two-way sampling using seasonal maxima approach for tropical and extra-tropical seasons. Colours denote the location (separated into East, Gulf, and West coast) and markers represent the different variable pairs. Black dots on markers indicate significant difference in dependence between seasons. Dashed lines show linear regression fits corresponding to all data points (black) and for different subsets according to locations (coloured as outlined in the legend).**





## 5 Discussion

In this study we have assessed the compound flooding potential from all four flooding drivers along the CONUS coastline. The dependence analysis is conducted using Kendall's $\tau$ and block maxima (either annual or seasonal maxima) with a two-way conditional sampling between flooding drivers. The choice of block maxima was to avoid having to identify individual thresholds and declustering windows for all sites and variable pairs individually when implementing POT. Previous studies, e.g., Ward et al. (2018) and Camus et al. (2021), indicate that using block maxima versus POT does not affect the overall results from large-scale dependence analyses in the context of compound flooding. Camus et al. (2021) showed through a comprehensive sensitivity analysis that using annual maxima and different thresholds in a POT framework leads to comparable results; while dependence values tended to be higher in the annual maxima approach, the spatial distribution (which is what we are mostly interested in) of the dependence was the same in both methods. Here, we find sites of highest dependence between the different pairs of drivers to be in the Gulf of Mexico, southeast, and southwest coasts (Fig. 3). For the Gulf and East coast this is due to the occurrence of hurricanes and tropical storms (which was confirmed in the second objective focused on the seasonal analysis) especially for pairs of drivers conditioned on surge. Dependencies using Kendall's $\tau$ were consistent with past regional and global studies (e.g., Wahl et al. (2015) for surge-precipitation, Ward et al. (2018) for surge-discharge, and Marcos et al. (2019) for surge-waves).

As could be expected, the highest dependence is found between S_W as both are oceanographic drivers, but we also find that significant dependence for S_P is more prevalent than for S_Q (especially along the northeast coast), highlighting the fact that catchment characteristics (e.g., size, surface type, steepness, and antecedent moisture content), rainfall intensity and duration, and snowmelt play an important role and not all dependence between S and P translates to dependence between S and Q (Hendry et al., 2019; Bevacqua et al., 2020; Couasnon et al., 2020). Additionally, more sites have significant dependence for Q_P than for P_Q, especially along the northeast coast, highlighting that not all extreme precipitation events can occur with moderate or high discharge, but extreme discharge events can occur with moderate or high precipitation. This shows again that other mechanisms contribute to high discharge events other than precipitation.

We find more places with significant tail dependence $\chi$ (Fig. 4) than with significant Kendall's $\tau$ (Fig. 3), and this is likely a result of how the data is sampled. Kendall's $\tau$ analysis includes extreme conditions of one variable (sampled in a year or season) and anything from low to extreme for the other, while tail dependence $\chi$ assesses the probability for one variable to be extreme when the other is extreme. This is done based on all daily values which are not declustered and hence prolonged high values could introduce stronger tail dependence.

From the seasonal analysis we find that for the different pairs of variables the dependence is always higher in the Gulf of Mexico during the tropical season as compared to the extra-tropical season (Fig. 5 and Fig. 6). This is attributed to the occurrence of hurricanes and tropical storms where low pressure systems accompanied by strong winds elevate coastal water levels through storm surges and also produce high waves. When the storm systems travel further inland, they often cause extreme precipitation leading to pluvial flooding and high river discharge leading to fluvial flooding. In both cases the flooding





impacts can be worse if drainage is blocked at the river mouth/outfall due to elevated coastal water levels, as happened in Texas in 2017 during hurricane Harvey (Emanuel, 2017). Along parts of the US East coast we find higher dependence during

the tropical season for flooding pairs S_Q, Q_S, S_P, and P_S which is likely also a result of the occurrence of hurricanes but could also be attributed to convective weather systems including thunderstorms that favour the occurrence of coastal and inland extreme events as shown by Catto and Dowdy (2021). The latter found that those weather types are more frequent during summer (tropical season) in association with increased thermodynamic instability and heating. On the other hand, we find higher dependence during the extra-tropical season on the West coast, especially for pairs conditioned on storm surge (S_P,

S_Q, and S_W). Bromirski et al. (2017) studied storm surges along the pacific coast of North America and found that storm surges peak during winter (December-February) caused by low pressure systems and in turn linked to high rainfall events driven by atmospheric rivers that occur on the West coast during winter. In this part of the country, the landfall of low-pressure systems causing high surge associated with extreme rainfall events compounds the adverse impacts of coincident high surge and waves on sea cliffs. On the East coast, the stronger dependence between W_S during the extra-tropical season compared

to the tropical season can be attributed to stronger wind-sea and swell energy during winter. Zheng et al. (2016) studied the spatial and seasonal distribution of wind-sea and swell energy and found that for the northern hemisphere the peak is in winter (December-February) and the seasonal average wind speed reaches a maximum during that time.

From the seasonal tail dependence analysis we find that results are more aligned with the 1:1 line (Fig. 7) compared to the rank correlation analysis (Fig. 5), and some of the conclusions are reversed, but mostly for sites where dependence is weak. This

shows how using different methods based on different subsets of the data can lead to slightly different results and conclusions. In a recent study, Camus et al. (2021) showed that tail dependence coefficients between two drivers were strongly positively correlated with joint occurrences of the same drivers, which was not always the case for Kendall's $\tau$. This implies that tail dependence $\chi$ is not always positively correlated with Kendall's $\tau$, especially when both are calculated using different subsets of the data sample, and explains the discrepancies found between Fig. 6 and Fig. 8.

In comparing dependence structures derived from model and observational data we followed the methodology in Zscheischler et al. (2021). Results showed that for pairs S-Q and S-P the tail dependence derived from models is very similar to that derived from observations in the Gulf of Mexico. The models underestimate the tail dependence for S-Q along the East and West coast, which might be a result of water management not captured by the models. For S-P, the models overestimate dependence at some sites along the East coast and underestimate it at most sites on the West coast. This points to spatial variations of models'

performance in estimating tail dependence for S-P. Moreover, weather types driving extreme inland and coastal events were found to be different on the West and East coast by Catto and Dowdy (2021). Models overestimate the tail dependence between S-W in the Gulf and southeast coast (Fig. 10). These are the locations where hurricanes and tropical cyclones occur and the seasonal analysis confirmed that tail dependence was overestimated by models particularly during the tropical season from Virginia to the western Gulf of Mexico (Fig. 12). In a comprehensive analysis for Europe, Paprotny et al. (2020) also compared

dependence structures between observations and model hindcasts and found that on average the dependence between surge and discharge was underestimated. Dependence between surge and precipitation on the other hand was overestimated along



the North Sea and English Channel but strongly underestimated in southern Europe. This existence of strong spatial variation in the ability of models to reproduce dependence structures between drivers (in particular for surge and precipitation) is also confirmed by our analysis.

**6 Conclusions**

We have quantified, for the first time, the compound flooding potential that arises from the combination of storm surge, waves, precipitation, and river discharge along the CONUS coastline. Our first objective was to characterize and map the dependence between the four different compound flooding drivers and identify spatial patterns. We carried out the analysis at 35 sites, where long enough overlapping datasets were available for the different variables. From a geographic perspective, more sites

with significant dependence between the different drivers exist along the Gulf, southeast, and southwest coasts as compared to the northwest and northeast. From a flooding driver perspective, the highest dependence is found between surges and waves which are both oceanographic drivers, followed by surge and precipitation, surge and discharge, waves and precipitation, and waves and discharge.

Our second objective was to perform a seasonal dependence analysis (tropical vs extra-tropical season). We found higher

dependence between the different drivers during the tropical season in the Gulf of Mexico and parts of the East coast that are prone to tropical cyclone impacts, whereas dependence was stronger on the west coast during the extra-tropical season. Differences between seasons were larger when using two-way sampling and Kendall's $\tau$ as a measure of dependence compared to when assessing tail dependence $\chi$; the latter leads to more similar results for both seasons. Seasonal differences in the strength of dependence between the different flooding drivers show in which season certain areas are more likely to be affected

by compound flooding, which can be integrated into coastal management and flood risk mitigation efforts.

Our third objective was to compare the dependence structure of different combinations of flooding drivers using observation-based and model-based data, where all model data were derived with coherent forcing from the state-of-the-art ERA5 reanalysis. For S_Q and S_P and in the Gulf of Mexico both models and observations point to the same dependence structure in the tails of the joint distributions. Models overestimate the tail dependence between P_W in all sites. On the West coast,

models also underestimate dependence in the tails in S_Q, S_P, and S_W, which is also found along the East coast but in fewer places and with the exception of S_P that is overestimated on the East coast. The seasonal analysis shows that models reproduce the dependence structure better during the tropical season compared to the extra-tropical season for the whole CONUS coastline.

Importantly, our study focuses only on the hazard component of flood risk, hence assessing the potential of compound flooding

caused by at least one extreme driver. Our assumption is that severe impacts can occur when at least one of the drivers is extreme (Wahl et al., 2015; Zscheischler et al., 2018), but from an impacts perspective this may not necessarily be the case. However, identifying which combinations of drivers have relatively higher dependence (and during which time of the year) is an important first step which can help identifying areas which require more scrutiny. The results can also guide choices in

terms of which types of models are required and need to be coupled to capture the relevant interactions between the four
flooding drivers.

## Code availability

Data pre-processing, analysis and visualization were carried out in R programming language (R Core Team, 2020). The
following R packages were used: 'dataRetrieval' (De Cicco et al., 2018) and 'rnoaa' (Chamberlain et al., 2016) for data
retrieval; 'dplyr' (Wickham et al., 2020b), 'lubridate' (Spinu et al., 2020), and 'tidyr' (Wickham, 2020) for data pre-processing.
'extRemes' (Gilleland and Katz, 2016) and other routines for data analysis; and 'ggplot2' (Wickham et al., 2020a) and
'pheatmap' (Kolde, 2015) for visualization.

## Data availability

Observational sea-level data is available from NOAA (http://tidesandcurrents.noaa.gov/), wave hindcast from USACE
(http://frf.usace.army.mil/wis/), river discharge from USGS (https://waterdata.usgs.gov/nwis), and precipitation from NOAA
(https://www.ncdc.noaa.gov/ghcnd-data-access). For reanalysis data, ERA5 (https://doi.org/10.24381/cds.adbb2d47),
GloFAS-ERA5 (https://doi.org/10.24381/cds.a4fdd6b9), and CoDEC (https://doi.org/10.24381/cds.8c59054f) data are
available from the Copernicus Climate Change Service (C3S) Climate Data Store.

## Author contributions

AAN and TW conceived the study. AAN performed the analysis and wrote the first draft of the paper. IDH, PC, and MMR
participated in technical discussions. All authors revised and co-wrote the paper.

## Competing interests

The authors declare that they have no conflict of interest.

## Acknowledgements

We thank Jakob Zscheischler and Philippe Naveau for making their code available to apply the KL divergence method and
Sanne Muis for providing storm surge model data for our selected locations from the CoDEC dataset. This material is based
upon work supported by the National Science Foundation (NERC-NSF joint funding opportunity) under NSF Grant Number
1929382 (TW and AAN). PC and IDH's time on this research have been supported by the UK NERC grant CHANCE (grant
no. NE/S010262/1).



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
