# Peer review of "Assessing the dependence structure between oceanographic, fluvial, and pluvial flooding drivers along the United States coastline"

_Hydrology and Earth System Sciences, 2021_

## Author Comment (AC1)

Dear Editor,

We, the authors, would like to thank the reviewers for their positive, thoughtful, and constructive feedback and comments on our submission titled "Assessing the dependence structure between oceanographic, fluvial, and pluvial flooding drivers along the United States coastline" submitted to HESS by Ahmed A. Nasr et al.

We have revised the text carefully, provided replies to comments, and amended the manuscript according to replies to questions raised. We outline the changes made point by point below (answer in blue and new in the manuscript in red, anything to be deleted in manuscript is highlighted in yellow).

Regards,

The Authors

**REVIEWER 1**

This is a very interesting study with three explicit objectives: 1) assessing the dependence between four main flood drivers along the coastline of CONUS (surge, wave, rainfall and river discharge), 2) assessing the seasonal patterns of such dependence, and 3) comparing tail dependence detected from observational records and from the reanalysis data. Not only the idea is very interesting, but also the research design is robust and alternative approaches (i.e. POT vs block-maxima and rank vs tail dependance) have been well-discussed. This study and its outputs are very helpful for coastal compound flood hazard assessment and management, as address a very important challenge that compound flooding community is dealing with it. Appropriate characterization of exceedance probability (which is instrumental to risk assessment) is a difficult task when multiple variables are involved and such studies help the flood risk modelers in at least two ways. First, they will now where there is no significant dependence between flooding variables and make the simplifying assumption of independence between variables. Second, in case a significant dependence is detected what type of numerical modeling scheme can provide the most relevant information about the nonlinear interactions between interacting variables. The manuscript is very well-written and results are nicely presented. Thus, I recommend publication of this manuscript in HESS after a minor revision. My comments are mainly about necessary justification/explanations and some editorial. Please, see the details below.

We thank the reviewer for the constructive feedback and comments.

[R1C1] My main concern here is the time-scale of hydrologic variables. While wave and surge are obtained and analyzed on an hourly basis, accumulated daily precipitation depth is taken as a representative of extreme pluvial flooding driver (L167). I doubt that such characteristic of rainfall can be a good representative of extreme precipitation contributing to compound coastal flooding events. This is a relatively long time window that somehow dampens the density of rain storm and so provides irrelevant information for compound coastal flood modeling. Please, explain if such selection was due to lack of data or methodological reasons? Anyways, I didn't find it explicitly discussed in your manuscript.

We thank the reviewer for raising this comment. Precipitation data is only available as cumulative daily depths for long enough periods to be useful for our study. Long hourly or sub-hourly records would be required to obtain the precipitation intensity, storm duration and depth per storm. However, with the available information, precipitation intensity which is important and can lead to severe flash floods in

small catchments with high runoff coefficients (or imperviousness) cannot be calculated. As a justified approximation which has been used in previous studies (e.g., Camus et al (2021) and Wahl et al (2015)), cumulative daily precipitation depths are used as proxy for pluvial flooding. From a meteorological perspective it's also unlikely that weather systems causing convective short-duration rainfall events also lead to high storm surge or waves. While this does not mean that they can't be involved in causing compound flooding, we would not expect to find significant dependence.

We will make the following changes in manuscript:

**[Line 163-169]:** We use precipitation observations from the Global Historical Climatology Network Daily (GHCN-D) hosted by NOAA's National Centers for Environmental Information (NOAA-NCEI) (https://www.ncdc.noaa.gov/ghcnd-data-access). For data retrieval we used the R package 'rnoaa' (Chamberlain et al., 2016). GHCN-D contains precipitation and other climate data from more than 100,000 stations worldwide covering periods ranging from 1 year to over 175 years. We consider the accumulated daily precipitation depth (similar to Camus et al (2021) and Wahl et al (2015)) since higher frequency data is not available for long enough time periods to be useful for our continental scale analysis. We use data from rain gauges that are located closest to the selected tide gauges with at least 20 years of overlapping data; in 31 instances the closest precipitation gauges providing long records are found within a 30 km radius around the respective tide gauges, for the other 4 sites the distance is larger but always smaller than 60 km.

[R1C2] Also, for the two-way selection scheme, I guess 10 days is too long to justify coincidence/concurrence. It's been justified with regard to the lag-time of river flow, but I doubt in any of the selected systems the transition time of the peak river flow from the upper tributaries to the outlet would be more than 5-6 days, at most. Do you have any reference on this? I am aware of other studies with comparable window length (i.e. 7 days), but "river flow lag time" might not be necessarily sufficient to justify such long sampling window length. Also, this is not a similar process between all cases. For example, such time window with justification from river discharge lag time has nothing to do with sampling wave and surge that have occurred up to 10 days apart. Moreover, even if it justifies the 10 days lag time in Q-S case, it does not automatically justify pairing an extreme surge with a river flow that has happened up to 10 days later. Such unified 10day time window needs a much stronger justification.

For two way sampling, variable time windows for up to ±10 days were applied (Line 200-202); that means that we tried different window sizes for different pairs and locations varying the window size between 0 and ±10 days and selected the one that maximizes Kendall's tau; i.e., we did not use a unified ±10-day time window.

Time windows of up to ±10 days were chosen to account for lag times (which we did not calculate) in addition to a number of days as buffer (concurrently or in close succession). The previous studies referred to by the reviewer where time windows of ±7 days were used probably include Ganguli and Merz (2019a;2019b) and Ganguli et al (2020). In these studies, first the discharge time series was lagged by a number of days calculated as a function of the catchment area, then the two-way sampling was performed between annual maxima surge and maximum value of discharge from lagged discharge time series within a time window of ±7 days.

The reason for the choice of a time window is that compound events do not have to occur concurrently, they can occur separated by a number of days. As stated in Ganguli et al (2020) "The choice of a window is motivated by the fact that two events do not need to occur at exactly the same day to enhance their

impacts. For instance, the first event could weaken flood defense systems or stress disaster management capacities, posing a problem when the second event occurs with a few days delay".

Regarding Surge and Waves, only values occurring within the same day were paired together. A window size of ±1 day was tested but results did not change as the highest values tend to occur on the same day.

The mean window size for different pairs ranged between 0 and ±5 days while the median ranged between 0 and ±4 days; only in very rare cases for certain pairs and in few locations window sizes of 10 days were considered for the final analysis. In the vast majority of cases much shorter windows were used. The below table shows the 10[th] percentile, mean, median, and 90[th] percentile of the time windows for different pairs.

| | S-Q | S-P | S-W | Q-P | Q-W | P-W |
|---|---|---|---|---|---|---|
| P10 (days) | 1 | 1 | 0 | 1 | 1 | 1 |
| Median (days) | 4 | 2 | 0 | 3 | 3 | 2 |
| Mean (days | 5 | 3 | 0 | 4 | 4 | 3 |
| P90 (days) | 10 | 5 | 0 | 6 | 7 | 5 |

We will make the following changes in the manuscript:

**[Line 200-204]:** maximum value of the other (conditioned) variable within a time window that we vary between 0 to ±10 days; the relatively long lag-times are chosen as we do not correct for the travel time of river flow from where it is measured/modelled and the tide gauge further downstream. Previous studies calculated these lag-times for river discharge and then applied a time window of ±7 days for sampling between surge and lagged discharge (e.g., Ganguli and Merz, 2019a; 2019b; Ganguli et al., 2020). From all the time windows tested we chose the one that maximizes dependence (Kendall's τ). For surge and wave, for example, that leads to the selection of a time window of 0 days as the high values usually occur on the same day. Across all locations and variable pairs the median time window that is selected is 3 days; window lengths of 10 days are only considered in rare cases for the S-Q combination. In general, the reason for the choice of a time window is that compound events do not have to occur on the same day to enhance impacts, they can occur separated by a number of days. The occurrence of one event could impact flood defense systems or disaster management efforts leading to enhanced impacts when another event occurs shortly after (Ganguli et al., 2020). The six variable pairs (and 12 combinations from the two-way sampling) are the following; the variable that is listed first if the conditioning variable (e.g., S_Q means that annual maxima S is paired with (near-) coincident Q, whereas Q_S means that annual maxima Q is paired with (near-)coincident S):

We will add the following references to the references list:

Ganguli, P. and Merz, B.: extreme coastal Water Levels exacerbate fluvial flood Hazards in northwestern europe, Scient. Rep., 9, 1–14, https://doi.org/10.1038/s41598-019-49822-6, 2019a.

Ganguli, P. and Merz, B.: Trends in Compound Flooding in Northwestern Europe During 1901–2014, Geophys. Res. Lett., 46, 10810–10820, https://doi.org/10.1029/2019GL084220, 2019b.

Ganguli, P., Paprotny, D., Hasan, M., Güntner, A., & Merz, B.: Projected changes in compound flood hazard from riverine and coastal floods in northwestern Europe, Earths Future, 8, https://doi.org/10.1029/2020EF001752, 2020.

[R1C3] I understand the value of information presented in Figures 5, 7, 9, but I don't see much value in the way that these are currently presented. There are simply too much information on a single chart. I suggest either breaking these down to subpanels (a, b, c, …) that separately plots the pairs, or removing these charts (as kind of similar information is already presented in the heatmaps). Moreover, I found the dashed lines simply meaningless here. Both rank and tail dependence measures used here assess the nonlinear dependencies and I don't understand what a linear regression means here? Please, explain if I am missing an important meaning behind these linear regression practice.

We thank the reviewer for this comment. The purpose of Figures 5,7,9, and 11 is to compare different statistics ($\tau$ or $\chi$) between seasons (Fig 5 and 7) or between observations and models (Fig 9 and 11). If there were no variations all marker points would lie on the diagonal (1:1) line. However, since there are variations, the marker points are scattered. The black dashed line represents the linear regression line between tropical and extra-tropical season (Fig 5 and 7) and between observations and models (Fig 9 and 11) for all variable pairs in all regions. The colored dashed lines show the same but for regions split into East, Gulf, and West coast. Although the rank correlation assesses the non-linear dependence, we use a linear dependence/correlation to assess the association between the rank correlation values obtained in different seasons (Fig 5 and 7) or using observations and models (Fig 9 and 11). We tested using different panels for the different regions but then we lose the information from looking at all regions combined; we also tested having 6 panels for each of the variable pairs, but that resulted in many large figures (12 panels would be needed for Fig. 11) without displaying more information.

The heatmaps in Figures 6 and 8 show the difference at individual locations in tropical and extra-tropical seasons. On the other hand, Figures 10 and 12 show the difference at individual locations in models and observations. Consequently, by removing Figures 5, 7, 9, and 11 (as suggested by the reviewer) information on the absolute values of dependence and tail dependence would be lost.

[R1C4] L142: WAM has not been defined yet.

We thank the reviewer. We will update the manuscript to be as follows:

**[Line141-142]:** We compare the WIS data against wave time-series extracted from the ERA5 reanalysis (spatial resolution = 0.5°×0.5° and temporal resolution = 1 hour) (Hersbach et al., 2020) based on the wave model WAM model (WAMDI Group, 1988).

We will add the following reference to the references list in the manuscript:

WAMDI Group: The WAM model—A third generation ocean wave prediction model, J. Phys. Oceanogr., 18, 1775-1810, https://doi.org/10.1175/1520-0485(1988)018<1775:TWMTGO>2.0.CO;2 , 1988

[R1C5] L224: The equation does not seem right to me. There should be a typo in it.

We thank the reviewer and we agree that there were missing brackets and these have been added. In the new form the equation should be correct (see also section 3.1 of Zscheischler et al (2021); https://doi.org/10.5194/esd-12-1-2021)

$$\chi = \lim_{q\to 1} P\left(F_a(A) > q | F_b(B) > q\right) \epsilon\ (0,1]$$

Reference:

Zscheischler, J., Naveau, P., Martius, O., Engelke, S., and Raible, C. C.: Evaluating the dependence structure of compound precipitation and wind speed extremes, Earth Syst. Dynam., 12, 1–16, https://doi.org/10.5194/esd-12-1-2021, 2021.

[R1C6] L305-321: Better to clearly explain earlier that numbers inside parenthesis show the number of sites.

We thank the reviewer for this comment. We do show that the numbers in the parenthesis refer to the number of sites in Line 305 in the manuscript. Additionally, we have added the word "sites" to each number and will update the manuscript as follows.

**[Line 305-320]:** discharge, out of 23 sites analysed, more sites show significant correlation for Q_S (14 sites) than S_Q (11 sites). Along the coasts of Florida and US southeast higher values of τ are found for S_Q than Q_S. In contrast, along the coasts in the western Gulf of Mexico and US southwest, the values of Q_S are higher than for S_Q. For surge and precipitation, from the 35 sites analysed, more sites show significant dependence in S_P (24 sites) compared to P_S (16 sites). Along the East coast, more sites possess significant dependence in S_P (14 sites compared to 9 sites in P_S) and at the same time the dependence values for S_P are higher than P_S values. Interestingly, along the Gulf and West coasts, although more sites have significant dependence in S_P (10 sites compared to 7 sites in P_S), the strength of the dependence is higher for P_S in most cases (6 out of the 7 sites); this is in agreement with results from Wahl et al. (2015). For surge and waves, out of 31 sites analysed, we find more sites with significant dependence in the case of S_W (25 sites) compared to W_S (16 sites), especially along the East coast. For the East and Gulf coasts, the strength of dependence is also higher for S_W compared to W_S, which is reversed on the West coast. For discharge and precipitation, out of 23 sites analysed, more sites show significant dependence in the case of Q_P (17 sites) compared to P_Q (13 sites), which is again most pronounced on the East coast. The strength of dependence of Q_P and P_Q in the Gulf and West coasts is higher than that for the East Coast. In most sites, the strength of dependence is higher for Q_P than P_Q. For waves and discharge, out of 17 sites analysed, only three show significant dependence in both cases. For W_Q all three are located in Florida and show relative high dependence strength. Lastly, for wave and precipitation, out of 31 analysed sites, there is approximately an equal number of sites showing significant dependence for W_P (12 sites) and P_W (11 sites). However, the strength of dependence is overall higher for W_P compared to P_W at most sites (4 of 5 sites) where both are significant.

[R1C7] L351: Looks like R here represents the Pearson corr coeff. (As mentioned in L441 and L413). If so please clarify first time appears in the manuscript.

We thank the reviewer and we agree and have made the following changes in the manuscript:

**[Line 212]:** correlation coefficient (R)

**[Line 351]:** Pearson's correlation coefficient R

[R1C8] L475-479: The sentence is too long. Please, consider breaking it down.

We agree with the reviewer and will make the following changes in the manuscript:

**[L475-479]:** As could be expected, the highest dependence is found between S_W as both are oceanographic drivers, but we also find that significant dependence for S_P is more prevalent than for S_Q (especially along the northeast coast)., highlighting This highlights the fact that catchment characteristics (e.g., size, surface type, steepness, and antecedent moisture content), rainfall intensity and duration, and snowmelt play an important role and not all dependence between S and P translates to dependence between S and Q (Hendry et al., 2019; Bevacqua et al., 2020; Couasnon et al., 2020).

[R1C9] L556: These citations are not necessary (better to say useful) at the end of conclusions.

We thank the reviewer and agree with this comment. We will remove them from the updated version of the manuscript as follows:

**[L555-556]:** caused by at least one extreme driver. Our assumption is that severe impacts can occur when at least one of the drivers is extreme (Wahl et al., 2015; Zscheischler et al., 2018), but from an impacts perspective this may not necessarily be the case.

Great work!

Thank you!

**REVIEWER 2**

Congrats to the authors for putting together an interesting and generally well-written manuscript. The study uses observational and model-based data to characterize the pair-wise dependence between various flood drivers (storm surge, waves, rainfall, river discharge) for sites across the entire CONUS and for different seasons. The authors then compare model-based dependence against the observed dependence structures between flood drivers to investigate the ability of models to represent multi-hazard dependence. I recommend publication after addressing some points I raise below.

We thank the reviewer for the constructive feedback and comments.

Main Points:

[R2C1] There are significant differences between the Kendall tau and tail dependence for the seasonal analysis. In particular, the kendall tau analysis suggests that the correlation is higher during the tropical season compared to extra tropical season (for the majority of locations and pairs). But the tail dependence analysis basically shows the exact opposite (figure 8 note all the blue boxes compared to the prevalence of red boxes in figure 6). The authors mention that some differences could be due to sampling differences and autocorrelation of the time series used for the tail dependence analysis, but I don't think this fully explains the large discrepancy between the two methods. Moreover, the authors do not comment on how to interpret these differences. Which metric should we use to quantify hazard dependence?

We thank the reviewer for this comment. For the Kendall $\tau$ analysis we performed a two-way sampling approach where we used extreme values (sampled using block (annual or seasonal) maxima approach) of one driver and the maximum values of the other driver within a time window. In that case, the other driver being chosen within a time window may or may not be extreme. On the other hand, for extremal (tail) dependance, using the daily timeseries (full or split into seasons), it calculates the probability of one driver being extreme (exceeding a threshold = 0.9) given that the other driver is extreme. The events here are not declustered or independent.

For two drivers (say A and B), the two-way sampling approach for Kendall's $\tau$ ensures that at least one of the drivers is extreme (that is A OR B are extreme). Extreme is defined as the annual (or seasonal) maxima value. On the other hand, for the tail dependence, it calculates the probability of one variable (say A) being extreme given that the other variable (say B) is extreme (directly proportional to A AND B are extreme). Extreme is defined as both values exceeding a threshold of 0.9.

Since both approaches use different samples from data and have different definitions, differences will inevitably occur. The tail dependence only characterizes compound events when both drivers are extreme (both exceed a certain threshold). On the other hand, Kendall's $\tau$ rank correlation analysis characterizes compound events generated when one of the drivers is extreme but not necessarily the other, providing information about the relative severity of the secondary driver. That's why we mentioned that both metrics provide insight on the dependence.

Considering two drivers, according to Wahl et al. (2015), compound flooding can occur when (1) two drivers are extreme (joint occurrence of two extreme events which is positively correlated to tail

dependence as in Camus et al. (2021)) or (2) when one of the drivers is extreme but not necessarily the other (at least one of the drivers is extreme which is covered by the two-way sampling)

The calculation of dependence using Kendall's τ and the two-way sampling is important for copula modeling at a later stage and obtaining design values (which is beyond the scope of this paper and was extensively applied in previous studies, see e.g.: Wahl et al (2015)). On the other hand, the strength of using tail dependence is seen in objective 3 of this study where we investigate the ability of models in reproducing the extremal dependence structure between variables and whether there is significant underestimation or overestimation according to KL divergence (see original manuscript Section 3.3, Lines 260-264).

Lastly, we will also provide more explanation in the discussion as follows (original manuscript section 5, lines [510-514]).

**lines [510-514]:** This shows how using different methods based on different subsets of the data can lead to slightly different results and conclusions, introducing subjectivity. In a recent study, Camus et al. (2021) showed that tail dependence coefficients between two drivers were strongly positively correlated with joint occurrences of the same drivers, which was not always the case for Kendall's τ. This implies that tail dependence χ is not always positively correlated with Kendall's τ, especially when both are calculated using different subsets of the data sample, and explains the discrepancies found between Fig. 6 and Fig. 8. The method of choice depends on the objective of the analysis. Deriving Kendall's τ is often an important interim step when performing joint probability analysis, e.g., using copula models. Tail dependence χ, on the other hand, is a very useful metric when assessing tail dependence structures, as done here for example when comparing model results and observations.

[R2C2] I didn't see the KL divergence results presented anywhere in the manuscript. As the authors point out the KL metric takes into account the entire dependence structure rather than just the tail dependence, so I was expecting to see/read the KL values for different locations and pairs. I think this needs to be added to the manuscript

We thank the reviewer for this comment. The KL calculations (the statistic $d_{12}$) on its own is a measure of distance or dissimilarity. According to Line 292-293: "The statistic $d_{12}$ follows a $\chi^2$ (W-1) distribution in the limit as the sample size approaches ∞ under suitable assumptions allowing us to estimate whether it is significantly different from zero.". Under this assumption, the KL divergence value ($d_{12}$) is significant if it lies outside the 95 % quantile of the limiting $\chi^2$ (W-1) distribution of the statistic $d_{12}$ under the null hypothesis of equal tail dependence structures (leading to rejection of the null hypothesis). This is reflected in Figure 9 and Figure 11 by black dots in the markers and in Figure 10 and Figure 12 by showing values of the cells (insignificant values are shown by an asterisk *). We noticed that captions for Figure 9,10,11, and 12 were not correct and we will update them in the new version of the manuscript (see reply for R2C11 and R2C12 for corrected captions). We debated whether to add all KL values (the statistic $d_{12}$) and the 95% quantile of the limiting $\chi^2$ (W-1) distribution of the statistic $d_{12}$ for all pairs and locations. That leads to a very large table which would have to go into a supplementary, which we don't have at the moment. After some discussions we decided against it as we believe that these values won't be of interest to the vast majority of the readers (and the final conclusions in terms of dependence structures being significantly different or not are already included in the figures). However, the table exists and we are happy to add it to a new supplementary material if the reviewer feels strongly about it.

[R2C3] I don't think the discussion could be organized better. Since one of the objectives/novelties of this work is to investigate hazard correlation across the entire CONUS, the discussion should be used to highlight regional trends. I suggest devoting one paragraph for each region, highlighting the main hazard correlations (and the meteorological reasons for the correlation), and then describing which season multi-hazard events are more likely to occur and why. This should be followed by a paragraph discussing the differences between the tail dependence and kendall tau analysis (see the first bullet). Then I would have a paragraph describing the comparison of the model with the observations. There are a lot of results presented in this paper so the discussion really needs to summarize everything in an organized manner.

The first sentence of this comment (saying that the discussion could NOT have been organized better) and the rest of it were contradictory and we assume that "don't" in the first sentence was a typo and interpreted it accordingly. We decided to organize the discussion according to the three objectives that we define in the introduction and highlight/discuss our key findings for each of these objectives in turn. While the first one is indeed purely focused on identifying spatial patterns, the other two are not, although the results of course also show spatial variability. We believe that structuring the discussion according to the objectives allows the reader to easily capture the main take-aways regarding these objectives. After reading it again we realized that by making some modifications to the text we can make the structure clearer, which will hopefully make it more accessible to the reader. We show the proposed changes to the discussion section below.

[revised manuscript text omitted]

[R2C4] I would also suggest adding some discussion about the limitations of using a bivariate approach for this analysis. For example, along the west coast of Florida, there is significant correlation between almost all the hazard pairs (Figure 3), suggesting that there may also be a significant threat for three or even all four of the hazards to occur simultaneously.

We thank the reviewer for the suggestion. We will add the below paragraph addressing it in the updated manuscript at the end of the discussion section (after Line 529).

In this study, we carried out bivariate dependence analysis between four main drivers which can potentially cause compound flooding. At some locations on the Gulf coast (e.g. west coast of Florida), significant correlation was found among most pairs of drivers. These are locations exposed to hurricanes and storms that can cause three or all four flooding drivers to coincide. The bivariate dependence analysis

presented here could be extended to include multivariate dependence, which can be modelled using higher dimension copulas (e.g. Bevacqua et al. (2017) and Jane et al (2020)).

We thank the reviewer for this comment. In line (306), the distinction of the southeast coast and Florida from the west gulf coast and southwest coast are mentioned for S-Q cases. The southeast region from station 15 to station 19 is only 5 stations and not all show similar behavior to the Gulf. The northeast region, on the other hand, contains 14 stations (station 1 to 14).

We produced a Figure as the reviewer suggested, but believe it does not add important new insights, while at the same time it would make the scatter plots throughout the manuscript more complex, as an additional color would have to be added.

We mentioned in [L322-323] that the Gulf, Southeast, and Southwest coasts are regions with higher and significant dependence compared to the northwest and northeast coasts. We will make sure in the revised version that differences between the southeast and northeast are highlighted in the text where appropriate and for cases where these differences are prominent (which is not the case for all of them, in some instances results are more comparable between the southeast and northeast).

[Figure]

*Figure 3: Dependence between different pairs of flooding drivers based on Kendall's τ and two-way sampling using annual maxima. Sites are grouped into Northeast, Southeast, Gulf, and West coast locations (see colours on the left and legend). The blue colour bar denotes dependence strength, blank squares indicate that data for the particular pair didn't exist or that the number of overlapping years was less than 20 and squares with * indicate that correlation is not significant.*

Section 4.2:

[R2C8] It doesn't really make sense to talk about a tropical cyclone season for the West coast. I think it's valid to look at the seasonal differences between dependence values for locations in the West coast, but the authors should provide some justification for why June-November vs December-May still makes sense since TCs do not impact the West coast.

We thank the reviewer for this comment. Indeed, we are looking at seasonal differences between June-November (referred to as Tropical season) vs December-May (Extra-tropical season). In the results in section 4.2 we show that there is higher dependence and tail dependence during December-May (extra-tropical season) vs June-November (tropical season) (Line 368, 384-385). We further provide an explanation in the discussion [ Section 5 Lines 499-502] by linking to meteorological conditions during

winter that favor the occurrence of coastal and inland flooding driven by high surge and atmospheric rivers, respectively.

We will implement the following changes in the manuscript:

**[Line 344-347]:** This section describes the results for the second objective, relating to the seasonal dependence analysis between the four drivers. Here we analyse data from the tropical cyclone season (June-November) June-November (tropical cyclone season for the Atlantic and Gulf coast) separately from the extra-tropical season (December-May) December-May (extra-tropical season) and compare results. First, we show the results from Kendall's rank correlation analysis applied to the two-way samples derived with the seasonal maxima method, and then we show results from analysing tail dependence (using daily timeseries split into seasons).

[R2C9] Figure 5: Again, I think it would be more interesting to show the southeast and northeast regions separately.

Please see reply on comment for Figure 3 [R2C7]. We have generated a new figure based on the suggestion of the reviewer. Both the Northeast and Southeast have the same regression line. We think that it will be too much information to have 4 regions distinguished in these scatter plots. A comment from Reviewer 1 indicated that these figures already contain a lot of information and breaking down regions even further would just add additional complexity. As outlined in our response above we will make sure to highlight differences between the southeast and northeast in the text in instances where these differences are prominent.

[Figure]

*Figure 5: Scatter plot comparing dependence derived with Kendall's τ and two-way sampling using seasonal maxima approach for tropical and extra-tropical seasons. Colours denote the location (separated into Northeast, Southeast, Gulf, and West coast) and markers represent the different variable pairs. Black dots on markers indicate significant difference in dependence between seasons. Dashed lines show linear regression fits corresponding to all data points (black) and for different subsets according to locations (coloured as outlined in the legend).*

Section 4.3:

[R2C10] Where exactly were the KL results presented? The authors mention that the KL divergence provides information about whether the dependence structure is significantly different between observations and model, but I did not find any figure/table presenting the KL results.

This comment relates to comment [R2C2], please see our response to that.

[R2C11] Figure 10: The caption here is not correct. The caption refers to differences between seasons, but I believe this figure is supposed to be showing the difference between reanalysis and observations

We thank the reviewer and agree that the caption was incorrect. We also found that captions for Figures 9 and 11 were not correct and this will be fixed in the revised manuscript as follows:

**[Line428-432]:** Figure 9: Scatter plot comparing extremal (tail) dependence (for q=0.9) derived using observations (x-axis) and models (y-axis) using daily time series of both variables. Colours denote the location (separated into East, Gulf, and West coast) and markers represent the different variable pairs. Black dots on markers indicate significant difference in tail dependence structure between observations and models according to KL divergence. Dashed lines show linear regression fits corresponding to all data points (black) and for different subsets according to locations (coloured as outlined in the legend).

**[Line434-438]:** Figure 10: Heat map showing differences in tail dependence (for q=0.9) derived for models (reanalysis) and observations using daily time series of both variables. Sites are grouped into East, Gulf, and West coast locations (see colours on the left and legend). The colour bar denotes the difference between χ in the reanalysis versus observations, where red colour denotes higher dependence in the reanalysis and blue colour denotes higher dependence in the observations. Squares with * indicate that difference in dependence is not significant. Blank squares indicate that data for the particular pair did not exist or that the number of overlapping years was less than 20.

**[Line449-453]:** Figure 11: Scatter plot comparing extremal (tail) dependence (for q=0.9) derived using observations (x-axis) and models (y-axis) for tropical (left) and extra-tropical (right) seasons using daily time series of both variables. Colours denote the location (separated into East, Gulf, and West coast) and markers represent the different variable pairs. Black dots on markers indicate significant difference in tail dependence structure between observations and models according to KL divergence. Dashed lines show linear regression fits corresponding to all data points (black) and for different subsets according to locations (coloured as outlined in the legend).

[R2C12] Figure 12: Caption incorrect again.

We thank the reviewer and agree that the caption was incorrect. We will correct it to the following:

**[Line455-459]:** Figure 12:  Heat map showing differences in extremal (tail) dependence (for q=0.9) derived using observations and models using daily time series of both variables for tropical (top) and extra-tropical (bottom) season. Sites are grouped into East, Gulf, and West coast locations (see colours on the left and legend). The colour bar denotes the difference between χ in the models versus observations, where red colour denotes higher dependence in the models and blue colour denotes higher dependence in the observations. Squares with * indicate that difference in dependence between models and observations is not significant. Blank squares indicate that data for the particular pair did not exist or that the number of overlapping years was less than 20.

[R2C13] Line 480-481: I don't understand this sentence: "not all extreme precipitation events can occur with moderate or high discharge, but extreme discharge events can occur with moderate or high precipitation". I think you mean that high precipitation can occur in the absence of high river discharge, but high river discharge usually occurs simultaneously with high precipitation.

We thank the reviewer for this comment. We will change this sentence to be the following as suggested by the reviewer

**[Line480-481]:** highlighting that ==not all extreme precipitation events can occur with moderate or high discharge, but extreme discharge events can occur with moderate or high precipitation== extreme

precipitation can occur in the absence of moderate or extreme river discharge, but extreme river discharge usually occurs simultaneously with moderate or extreme precipitation